

# Mercury anomalies across the Palaeocene-Eocene Thermal Maximum

Morgan T. Jones[1], Lawrence M.E. Percival[2†], Ella W. Stokke[1], Joost Frieling[3], Tamsin A. Mather[2], Lars Riber[4], Brian A. Schubert[5], Bo Schultz[6], Christian Tegner[7], Sverre Planke[1,8], Henrik H. Svensen[1]

[1]Centre for Earth Evolution and Dynamics (CEED), University of Oslo, PO Box 1028 Blindern, 0315 Oslo, Norway
[2]Department of Earth Sciences, University of Oxford, South Parks Road, Oxford, OX1 3AN, UK
[3]Department of Earth Sciences, Utrecht University, Princetonlaan 8a, 3584 CB, Utrecht, Netherlands
[4]Department of Geosciences, University of Oslo, PO Box 1047 Blindern, 0316 Oslo, Norway
[5]School of Geosciences, University of Louisiana at Lafayette, 611 McKinley St, Hamilton Hall #323, Lafayette, LA 70504 USA
[6]Museum Salling - Fur Museum, 7884 Fur, Denmark
[7]Department of Geoscience, Aarhus University, Høegh-Guldbergs Gade 2, building 1672, 321, 8000 Aarhus, Denmark
[8]Volcanic Basin Petroleum Research (VBPR AS), Forskningsparken, Gaustadalléen 21, 0349 Oslo, Norway
[†]Current address: Institut des sciences de la Terre, Géopolis, Université de Lausanne, 1015 Lausanne, Switzerland

*Correspondence to*: Morgan T. Jones (m.t.jones@geo.uio.no)

**Abstract.** Large-scale magmatic events like the emplacement of the North Atlantic Igneous Province (NAIP) are often coincident with periods of extreme climate change such as the Palaeocene–Eocene Thermal Maximum (PETM). One proxy for volcanism in the geological record that is receiving increased attention is the use of mercury (Hg) anomalies. Volcanic eruptions are among the dominant natural sources of Hg to the environment; thus, elevated Hg/TOC values in the sedimentary rock record may reflect an increase in volcanic activity at the time of deposition. Here we focus on five continental shelf 20 sections located around the NAIP in the Paleogene. We measured Hg, total organic carbon (TOC) concentrations, and $\delta^{13}$C values to assess how Hg deposition fluctuated across the carbon isotope excursion (CIE). We find a huge variation in Hg anomalies between sites. The Grane field in the North Sea, the most proximal locality to the NAIP analyzed, shows Hg concentrations up to 90,100 ppb (Hg/TOC = 95,700 ppb/wt%) in the early Eocene. Significant Hg/TOC anomalies are also present in Danish (up to 324 ppb/wt%) and Svalbard (up to 257 ppb/wt%) sections prior to the onset of the PETM and during 25 the recovery period, while the Svalbard section also shows a continuous Hg/TOC anomaly during the body of the CIE. The combination with other tracers of volcanism, tephra layers and unradiogenic Os isotopes, at these localities suggests that the Hg/TOC anomalies reflect pulses of magmatic activity. In contrast, we do not observe clear Hg anomalies on the New Jersey shelf (Bass River) or the Arctic Ocean (Lomonosov Ridge). This large spatial variance could be due to more regional Hg deposition. One possibility is that phreatomagmatic eruptions and hydrothermal vent complexes formed during the 30 emplacement of sills led to submarine Hg release, which is observed to result in limited distribution in the modern. The Hg/TOC anomalies in strata deposited prior to the CIE may suggest that magmatism linked to the emplacement of the NAIP contributed to the initiation of the PETM. However, evidence for considerable volcanism in the form of numerous tephra layers and Hg/TOC anomalies post-PETM indicates a complicated relationship between LIP volcanism and climate. Factors such as



climate system feedbacks, changes to the NAIP emplacement style, and/or varying magma production rates may be key to both the onset and cessation of hyperthermal conditions during the PETM.

## 1 Introduction

The Palaeocene–Eocene Thermal Maximum (PETM; 55.8 Ma) was a rapid and prolonged global warming event (Charles et
al., 2011). It is marked by a sharp negative carbon isotope excursion (CIE) of 3–5 ‰ (McInerney and Wing, 2011), caused by the voluminous release of [13]C-depleted carbon to the ocean-atmosphere system (Dickens et al., 1995; Zachos et al., 2008). This carbon cycle perturbation and associated global warming (4–5 ºC; Dunkley-Jones et al., 2013; Frieling et al., 2017) instigated changes to both the marine and terrestrial realms (Dickson et al., 2012; Gingerich, 2006; Jaramillo et al., 2010; Zachos et al., 2005). The PETM CIE represents the highest magnitude example of several such excursions from Paleocene and
Eocene strata, but is also distinct from the all other Cenozoic hyperthermal events in terms of its long duration (Stap et al., 2009; Zeebe et al., 2009). Furthermore, the other Paleogene hyperthermals events are generally considered to be astronomically paced (Cramer et al., 2003; Lourens et al., 2005; Westerhold et al., 2009), while such a relation for the PETM is still debated. The PETM CIE in sedimentary records is interpreted as documenting a rapid "onset phase" to the event (~1–5 kyr; (Kirtland-Turner et al., 2017; Kirtland-Turner and Ridgwell, 2016; Zeebe et al., 2016), followed by a "body phase" marked by stable
but anomalously low $\delta^{13}$C values for around 70–100 kyr, and a "recovery phase" that took approximately 50–100 kyr (Murphy et al., 2010; Röhl et al., 2007).

Numerous carbon sources have been suggested, either in concert or individually, to explain the onset and duration of the extreme greenhouse conditions. These include the release of marine gas hydrates (Dickens et al., 1995; Zeebe, 2013),
permafrost thawing (DeConto et al., 2012), a bolide impact (Kent et al., 2003; Schaller et al., 2016), and direct and thermogenic degassing associated with large scale magmatism in the North Atlantic Igneous Province (NAIP) (Eldholm and Thomas, 1993; Gutjahr et al., 2017; Storey et al., 2007a; Svensen et al., 2004). It is challenging to directly link these [13]C depleted sources to the PETM in the stratigraphic record. The observed global warming in sedimentary records requires the release of at least 10,000 Gt of C in most state-of-the-art climate models (Frieling et al., 2017), which is greater than any of the estimated fluxes
from the above hypothesized sources can solely account for (Aarnes et al., 2015; DeConto et al., 2012; Gu et al., 2011; Gutjahr et al., 2017; Zeebe et al., 2009). Furthermore, several studies have indicated that initial warming leads the CIE by several thousand years, suggesting that a positive feedback triggered the release of a [13]C-depleted source (Dickens, 2011; Frieling et al., 2018; Kender et al., 2012; Sluijs et al., 2007b). This early warming is not associated with a negative shift in $\delta^{13}$C values, which indicates that there was either an astronomical configuration favorable to induce (high-latitude) warming (DeConto et
al., 2012; Lunt et al., 2011), or that the initial warming was driven by the release of a less [13]C-depleted source compared to emissions during the main PETM CIE.

The observation that global warming appears to begin before the onset of the CIE suggests that the emplacement of the NAIP is a mechanism worthy of particular attention. Large igneous provinces (LIPs) such as the NAIP are characterized by the voluminous and rapid emplacement of magma into the shallow crust and at the surface (Burgess and Bowring, 2015; Ernst, 2014). There is a strong positive correlation between the emplacement of LIPs and both rapid climate change events and mass

extinctions in Earth history (Courtillot and Renne, 2003; Wignall, 2001), suggesting a possible causal connection. The NAIP is one of the larger known LIPs (Ernst, 2014) and occurred in two main pulses (Wilkinson et al., 2016). The second and more voluminous pulse began at ~56 Ma, emplacing an estimated $5\text{-}10 \times 10^6$ km$^3$ of magma and lava within a few million years (Figure 1; Saunders, 2015; White et al., 1987) and is closely associated with the opening of the Northeast Atlantic (Saunders et al., 2007; Storey et al., 2007b; White and McKenzie, 1995). Substantiating a causal relationship between the NAIP and the

PETM is complicated by several factors, including limited geochronological data. While extinction and climate records may be readily apparent in sedimentary sections worldwide, it is uncommon to find measureable LIP products such as tephra layers in these sections due to transport limitations, and post-depositional factors such as reworking and weathering (Jones, 2015). This limits their use as potential marker horizons. Therefore, a well-tested and uniquely volcanic tracer in sedimentary rocks would be a powerful proxy for understanding the relationship between large-scale volcanism and rapid climate change events.

## 2 Mercury as a proxy for NAIP activity

Mercury (Hg) anomalies in sedimentary records have been proposed as an indicator for periods of major volcanic activity (Sanei et al., 2012). There is a strong positive correlation between Hg and total organic carbon (TOC) in modern sediments (Gehrke et al., 2009; Outridge et al., 2007; Ruiz and Tomiyasu, 2015; Sanei et al., 2014). Normalizing Hg to Hg/TOC helps to correct for variations in sedimentation rate, as TOC is generally the primary phase to complex Hg (Sanei et al., 2012).

Therefore, sharp increases in Hg/TOC ratios preserved in the sedimentary record are interpreted as a proxy for enhanced volcanic activity. Such anomalies have been reported as a tracer for volcanism for several major environmental perturbations and/or mass extinctions in the geological record (Font et al., 2016; Grasby et al., 2013; 2016; Keller et al., 2018; Percival et al., 2015; 2017; Sanei et al., 2012; Sial et al., 2013; Thibodeau et al., 2016). Therefore, this method is an important proxy to assess the relative importance of volcanism as a causal mechanism for the PETM, given the wealth of literature on the PETM

and the availability of numerous sedimentary sections worldwide. A recent pioneer study using the Paleocene-Eocene Global Boundary Stratotype Section and Point (GSSP) in Dababiya, Egypt, showed a strong Hg/TOC anomaly within the PETM CIE (Keller et al., 2018).

There are two principal mechanisms for the emplacement of the NAIP to affect the Hg cycle (Figure 2): 1) Effusive and

explosive volcanism, and 2) explosive hydrothermal vent complexes releasing gases generated by contact metamorphism of organic-rich sediments.



## 2.1 Volcanism

Present day volcanism is a well-established flux of Hg to the atmosphere (Pyle and Mather, 2003), and the NAIP was particularly active in the late Paleocene and early Eocene (Larsen et al., 1999; Storey et al., 2007b). In East Greenland, studies using fluid inclusions and amphiboles as geobarometers suggested that the Skaergaard intrusion was buried by a huge

outpouring of continental flood basalts (5.3–6.3 ±2.7 km) during the voluminous second phase of NAIP as the magma chamber crystallized (Larsen and Tegner, 2006). High precision U-Pb ages indicate an initial emplacement at ca. 56.02 Ma for the intrusion (Wotzlaw et al., 2012). The predicted timescale of cooling and therefore the emplacement of the overlying enormous volumes of flood basalts, is around 100–300 kyr (Larsen and Tegner, 2006). The U-Pb ages in East Greenland at the onset of widespread flood volcanism predate the onset of the PETM by about 200 kyr (Charles et al., 2011). A distinctive tephra layer

named the Gronau tuff has been identified in the uppermost part of the East Greenland flood basalts (Heister et al., 2001), which is indistinguishable from tephra layer #-17 from the Fur Formation (Denmark) in both chemistry and age (Storey et al., 2007a). The corrected Ar/Ar radiometric age, using the most recent Fish Canyon Tuff correlation (Kuiper et al., 2008), is calculated to be 55.6 ± 0.1 Ma for tephra layer #-17. This indicates that the voluminous flood basalt volcanism in Greenland was emplaced in a 400 kyr time interval that encompasses the PETM.

Any Hg/TOC anomalies resulting from flood basalt province emplacement will be affected by the style and frequency of eruptions (Figure 2). Pulsed activity with considerable repose periods between eruptions might be expected to result in sharp individual Hg/TOC peaks, while continuous eruptions could result in a sustained Hg/TOC anomaly, depending on data resolution and the sedimentary record in question. The explosiveness of eruptions will also dictate the height to which Hg is

20 emitted in the atmosphere, which can affect the regional to global transport of Hg. There are widespread early Eocene tholeiitic basalt tephras from the NAIP preserved in the Balder Formation in the North Sea (Haaland et al., 2000) and in the upper Fur Formation in Denmark (Bøggild, 1918; Larsen et al., 2003), which are interpreted to be formed by shallow marine, Surtseyan-type eruptions (Planke et al., 2000). Phreatomagmatic eruptions may transfer part of the erupted Hg to seawater, resulting in more limited global transport (Figure 2). The modern residence time of Hg in oceanic waters is decades to centuries (Gill and

25 Fitzgerald, 1988; Gworek et al., 2016), which is shorter than the mixing time of the ocean, so any Hg released to the water is likely to be redeposited in closer proximity to the source compared to atmospheric emissions (Scaife et al., 2017).

## 2.2 Contact Metamorphism

There is mounting evidence that magmatic intrusions during the emplacement of the NAIP led to the release of huge volumes of greenhouse gases (Figure 2). Thousands of submarine hydrothermal vent complexes have been identified by seismic

reflection profiles from the offshore Vøring and Møre basins in the Norwegian Sea (Svensen et al., 2004). Vent structures form at the edges of sill intrusions due to overpressure generated by boiling of pore fluids and/or gas generation during contact metamorphism (Aarnes et al., 2015; Iyer et al., 2017). The resulting explosions are capable of ejecting gases into the





atmosphere, even from submarine vents (Svensen et al., 2004). Hydrothermal vent complexes around large sill intrusions have also been observed in the Faroe-Shetland basin (Hansen, 2006) and on the northeast Greenland margin (Reynolds et al., 2017) (Figure 1). It is therefore likely that these features were widespread along the proto-northeast Atlantic margins. Around 95% of the hydrothermal vent complexes in the Vøring and Møre basins terminate at the horizon between Paleocene and Eocene

strata, with the remainder terminating within the Paleocene sequence (Planke et al., 2005; Svensen et al., 2004). One zircon U-Pb age from a sill in the Vøring margin was dated to 55.6 Ma (Svensen et al., 2010) and it was recently shown that the only drilled hydrothermal vent complex dates to within the body of the CIE (Frieling et al., 2016). It therefore appears likely that there was considerable emplacement of sills with associated hydrothermal venting of gases along the margins of the NAIP coincident with the PETM, although the activity in these widespread vent systems likely encompasses a more prolonged

interval of time (Kjoberg et al., 2017; Svensen et al., 2004).

Sill intrusions are likely to be a source of Hg to the environment, both from magmatic degassing and the volatilization of organic material during contact metamorphism of intruded sedimentary rocks. The resulting Hg/TOC signal in the sedimentary record will be dependent on the timescales of thermogenic gas generation and eruption, which for individual sills is predicted

to be in the order of 100's to 10,000's years (Heimdal et al., 2018; Stordal et al., 2017). Vent structure appear to have been conduits of multiple gas ejections during their lifetime (Svensen et al., 2003), so gas release is likely to occur in pulsed fashion that would be closely temporally spaced. This would manifest as a larger initial Hg/TOC anomaly following a sill intrusion, followed by continuously elevated Hg/TOC values. As with the phreatomagmatic eruptions, the resulting Hg anomalies would likely be only regional in extent if a substantial part of the Hg was transferred to the water (Figure 2).

Here we explore the occurrence of Hg/TOC anomalies in PETM sediments from several shallow marine sections with varying proximity the NAIP. To elucidate the timing and potential causality, we compare our data to the CIE and warming and discuss the relevance in a carbon cycle and climate framework.

### 3 Materials and Methods

Five sites were investigated in this study (Figure 1). Four of these are drill cores through the PETM interval from the North Sea, Svalbard, New Jersey (USA), and the Arctic Ocean, and one is a field outcrop in Denmark. The localities were selected as they are continental shelf successions with little disruption to the stratigraphy, and the organic matter is well preserved with abundant biomarkers and fragile microfossils (Pagani et al., 2006; Schoon et al., 2015; Sluijs et al., 2007a), which minimizes the effects of syn- and post-depositional process on Hg/TOC signals. The sampling of each site is described in turn:



## 3.1 Grane Field, North Sea

The Grane field is situated in the Southern Viking Graben in the Norwegian North Sea (Figure 1). Four wells were drilled by Norsk Hydro Produksjon AS (now Equinor ASA) from 1993 to 1995, which have been the focus of previous investigations (Haaland et al., 2000; Mangerud et al., 1999). Two distinct phases of volcanism are recorded by tuffaceous intervals, phase

one in the Våle and Lista Formations (mid Paleocene), and phase two in the Sele and Balder Formations of the late Paleocene and early Eocene (Haaland et al., 2000). The Balder Formation is main tuffaceous interval with over 160 discreet ash layers (Haaland et al., 2000). Core 25/11-17 (59°3'27"N, 2°29'07"E) was sampled in 2008 by S. Polteau and A. Mazzini. The samples used here cover an interval between 1660 and 1643 meters below surface (m.b.s.). This encompasses the end of the Lista Formation, the Sele Formation, and the start of the Balder Formation (Figure 3). The dominant lithology is claystone in the

Lista Formation and siltstone in the Sele Formation. Occasional tephra layers are present in Lista Formation and are numerous in the Sele Formation. The Balder Formation (1652-1630 m.b.s. in this core) is the main tephra-rich interval that is documented across the North Sea. Two oil sands are found in core 25/11-17, at 1766.6 and 1783.7 m.b.s., respectively (Våle Formation).

## 3.2 Central Basin, Svalbard

The Paleogene Central Basin in Svalbard was drilled by Store Norske Spitsbergen Kulkompani (SNSK) at Urdkollen (core

BH09/2005; 77°54'09"N, 16°12'59"E; Figure 1). This core has been the focus of numerous previous investigations (Charles et al., 2011; Cui et al., 2011; Dypvik et al., 2011; Harding et al., 2011; Nagy et al., 2013; Riber, 2009; Wieczorek et al., 2013), allowing direct comparisons with published datasets. The Central Basin began to form as a rapidly subsiding foreland basin in the mid Paleocene (Jones et al., 2017). By the late Paleocene and early Eocene, the sedimentary environment was consistently marine deposition with varying degrees of terrigenous input. The shale-dominated Frysjaodden Formation begins ~20 m below

the onset of the PETM and continues up to 340 m above the CIE (Figure 4). The rapid deposition rate of 21.5–27.4 cm/kyr during the PETM (Charles et al., 2011), combined with little change in lithology across the interval and relatively high and consistent TOC concentrations, makes this expanded section ideal for studying Hg anomalies across the PETM. A total of 127 samples were collected for this study and from previous research (Riber, 2009), covering an interval from 554 to 470 m.b.s.

## 3.3 Fur Island, Denmark

The island of Fur, located within Limfjorden in northern Denmark (Figure 1), features outcrops dominated by the early Eocene Stolleklint Clay and the overlying clay-rich to clay-poor diatomite of the Fur Formation (Figures 5-6; Heilmann-Clausen et al., 1985; Pedersen et al., 2004; Pedersen and Surlyk, 1983). The ~60 m thick Fur Formation contains over 170 distinct tephra horizons that are interpreted to be derived from explosive NAIP volcanism (Larsen et al., 2003). Tephra layers are also present in the Stolleklint Clay but are less numerous. The ashes are separated into a negative and positive series (Bøggild, 1918), with

the boundary between the Stolleklint Clay and the Fur Formation placed at the prominent thick silicic ash layer #-33 (Heilmann-Clausen et al., 1985). The Paleocene–Eocene boundary is at the base of the Stolleklint Clay, which overlies the





Holmehus Formation and/or the Østerrende Clay (Heilmann-Clausen, 1995; 2006). The base of the Stolleklint Clay is rarely exposed at the surface on Fur Island, except at the type locality of Stolleklint Beach after the occasional extreme storm clears the beach sand (56°50'29"N, 8°59'33"E). One such brief exposure in 2005 has been the focus of several investigations (Heilmann-Clausen et al., 2014; Schoon et al., 2013; 2015).

Here we use outcrops at Stolleklint beach and a road-cut into a quarry to the southwest (56°49'51"N, 8°58'45"E) to sample the body and recovery of the PETM, using the tephra layers #-35, #-34, and #-33 as marker horizons to calibrate between sections. In order to evaluate the lower part of the Stolleklint Clay and underlying Holmehus-Østerrende Clay, we excavated a 43 m long and 0.5 m deep trench in the beach. We used numerous dip and strike measurements and trigonometry to obtain a

calculated local true thickness for the Stolleklint Clay of 24.4 m (maximum error ±2 m, see Supplementary data for details), from basal contact with the Holmehus-Østerrende Clays to the base of tephra #-33. This estimated thickness of the Stolleklint Clay is at the upper end of previous estimates (Schoon et al., 2013; 2015). Sub-beach samples were collected every 0.01 m across the boundary between the Holmehus-Østerrende Clay and the Stolleklint Clay, then every 0.5 m along the beach. This equates to a sampling interval of 0.2-0.3 m once converted through the trigonometry estimates. Additional samples were

collected along the Stolleklint cliff face. The quarry road-cut section was sampled every 0.1 m across a ~7m section below and above tephra layer #-33.

Three tephra layers are present at the base of the Stolleklint Clay, with the second layer coinciding with the onset of the CIE (Heilmann-Clausen et al., 2014; Figure 6). These layers are given the names 'SK1', 'SK2' and 'SK3' as they do not correspond

to any of the numbered ash series described by Bøggild (1918). Based on textural similarities and known distributions of NAIP tephras (Haaland et al., 2000; Westerhold et al., 2009), it is highly likely these ash layers originated from the NAIP. No other tephra layers have been identified in the underlying Paleocene strata (Holmehus Formation and Østerrende Clay) in this or any other locality in Denmark (Heilmann-Clausen et al., 2014).

### 3.4 Lomonosov Ridge, Arctic Ocean

The core 302-4A was collected during the Arctic Coring Expedition (ACEX) cruise to the Lomonosov Ridge, located at 87°52'12"N, 136°12'41"E (Figure 1) approximately 250 km from the North Pole (Backman et al., 2004). This section is well studied (Pagani et al., 2006; Sluijs et al., 2008; Sluijs et al., 2006; Waddell and Moore, 2008; Weller and Stein, 2008) and we use the same sample set as Sluijs et al. (2006) here. The lithostratigraphic unit during the interval between the late Paleocene and early Eocene are dark gray, fine-grained, siliciclastic-dominated clays that contain minor amounts of pyrite and biogenic

silica (Figure 7; Backman et al., 2004). Core recovery of PETM strata was poor, with only 55 cm disturbed core (302-31X) recovered from anywhere between 388 meters composite depth (m.c.d., the top of core 302-32X) and 384.54 m.c.d. (the bottom of core 302-30X; Sluijs et al., 2006).





### 3.5 Bass River, New Jersey

A core was collected from the ODP 174AX "Bass River" Leg in New Jersey (39°36'42"N, 74°26'12"W; Figure 1) that includes the Paleocene–Eocene boundary (Cramer et al., 1999). The site was located close to the edge of the American continental shelf in the late Paleocene (Miller et al., 2004). The sediments are characterized by glauconite-rich silts and sands that fine into clay-rich sediments just below and across the PETM CIE (John et al., 2008). A sandstone unit is found at the top of the CIE, with an unconformity separating it from continued claystone above (Figure 8). This site has been the focus of numerous investigations (Babila et al., 2016; John et al., 2008; Moore and Kurtz, 2008; O'Dea et al., 2014; Sluijs et al., 2007b; Stassen et al., 2012), and represents the most distal locality to the volcanic activity of the NAIP studied here. Samples were collected at 6-50 cm intervals over a 25 m section, with the lowest resolution in sediments recording the onset and recovery of the PETM due to severe core depletion of those strata.

### 3.6 Sample Analyses

Mercury analysis was conducted using the Lumex RA-915 Portable Mercury Analyzer with a PYRO-915 pyrolyzer (Bin et al., 2001) at the University of Oxford. Analytical procedures followed in-house protocols (Percival et al., 2017), where 40–60 mg of powdered sample was weighed before being transferred to the pyrolyzer and heated to 700 °C. Sample peaks in the spectrometer were calibrated using the NIMT/UOE/FM/001 peat standard with a known Hg concentration of 169 ±7 ppb, with repeat calibrations every ten samples to account for any drift in the instrument. Each sample was run in duplicate to further reduce the analytical error to around ±5 %.

Carbon isotope and total organic carbon (TOC) values used published data, where available (Charles et al., 2011; John et al., 2008; Sluijs et al., 2006). Additional TOC analyses were conducted for Svalbard and Fur samples on the Rock Eval 6 (Espitalié et al., 1977) at the University of Oxford. New carbon isotope analyses were conducted for Fur and Grane samples. Samples were weighed into tin capsules, then the $\delta^{13}C$ value of each sample was determined in triplicate using a Delta V Advantage Isotope Ratio Mass Spectrometer (Thermo Fisher, Bremen, Germany) configured to a Thermo Finnigan 1112 Series Flash Elemental Analyzer at the University of Louisiana at Lafayette. All data are reported in standard δ-notation (in units of permil, ‰) and normalized to the Vienna Pee Dee Belemnite (VPDB) scale using three internal lab reference materials (JRICE: $\delta^{13}C$ = -27.44‰; JHIST: $\delta^{13}C$ = -8.15‰; and JGLY: $\delta^{13}C$ = -43.51‰). A quality assurance sample (JGLUC) was analyzed as an unknown. Across all analyses, the JGLUC quality assurance sample averaged $\delta^{13}C$ = -10.64 ± 0.13 (n = 10), which is in agreement with the calibrated value of -10.52‰. For the sediment samples, the median standard deviation of the replicate capsules was 0.10‰. Combustion also resulted in a quantification of percent carbon (%C) in each sample.



## 4 Results

The full results are shown in Figures 3-8 for individual localities, and the raw data can be found in the Supplementary data. Data for the Dababiya Quarry GSSP locality (Keller et al., 2018) is also shown in Figure 9.

### 4.1 Grane Field, North Sea

The extent of the carbon isotope excursion (CIE) is difficult to determine in the Grane core (Figure 3). The body phase is apparently represented by only one data point (-30.6 ‰ at 1651.05 m.b.s.; Figure 3), which is directly followed by a return to values around -27 ‰, albeit more negative than pre-PETM conditions (-25 ‰). The shape and duration of the CIE in the Grane core is markedly different from other Sele Formation localities in the North Sea (Kemp et al., 2016; Kender et al., 2012), where $\delta^{13}C$ values remain close to -30 ‰ even after the recovery of the CIE. Therefore, while the onset of the PETM CIE is apparent,
it is difficult to unambiguously identify the body and recovery phases of the CIE in the stratigraphy.

The Hg concentrations throughout the studied section of core 25/11-17 (Figure 3) are considerably higher than any published Hg concentrations in the sedimentary rock record (Hg ≤ 90,100 ppb, Hg/TOC ≤ 95,700 ppb/wt%), despite relatively low TOC concentrations (≤1.85 wt%). For comparison, the extreme Hg concentrations measured across the Permian–Triassic boundary
in Svalbard and the Canadian Arctic peaked at 680 ppb (Grasby et al., 2016; Sanei et al., 2012). The lowest measured Hg concentration in the whole section is 209 ppb from a tuff in the Sele Formation, which would represent a significant Hg anomaly in any other section. The largest Hg peaks occur just before the base of the CIE in the lower Sele Formation (30,500 ppb) and in the sandy interval at the start of the Balder Formation above the PETM (90,100 ppb). Two oil sands much lower in the succession (1766.6 and 1783.7 m.b.s.) had Hg concentrations of 1135 and 3295 ppb, respectively (see supplementary
data).

### 4.2 Central Basin, Svalbard

The CIE of the core 09/2005 is unusual for a PETM section in that there is a prolonged (6.2 m thick) onset and a gradual recovery from negative $\delta^{13}C$ values (Figure 4) (Charles et al., 2011), rather than a continuous negative $\delta^{13}C$ excursion that clearly defines the body of the CIE in other records. Consequently, it is difficult to precisely define where the CIE body ends
and recovery begins. Two thin tephra layers of unknown provenance are found at 517 and 511 m.b.s. within this interval. There are two distinct types of Hg anomalies in this section (Figure 4). First, there are sporadic clusters of large but transient spikes in Hg/TOC ratios, particularly in the Paleocene strata at 552–548 m.b.s., prior to the onset of the CIE (540–534 m.b.s.), during the recovery phase of the PETM CIE (515–509 m.b.s.), and later in the Eocene (483–476 m.b.s.). The second observed type of anomaly is a sustained elevation in Hg concentrations that occurs prior to the onset (536–534 m.b.s.) and throughout the
main part of the CIE (530–509 m.b.s.; Figure 4). Mean Hg concentrations (63 ppb) and Hg/TOC ratios (51 ppb/wt%) are consistently elevated during the body of the CIE. For comparison, mean values during the recovery phase and early Eocene



(508–483 m.b.s.) are 15 ppb Hg and a Hg/TOC ratio of 12, despite little change in TOC concentrations or lithology (Figure 4).

### 4.3 Fur Island, Denmark

The results of the samples collected from the Stolleklint beach and road-cut localities are shown in Figures 5 and 6. The stratigraphic logs are calibrated so that the top of the tephra layer #-33 is zero. The onset of the CIE at Fur spans only 0.15 m (Figure 6), whereas the body and recovery of the CIE (Figure 5) span around 24.3 and 5 m, respectively. Coupled with a marked change from the heavily bioturbated, glauconite rich Østerrende/Holmehus Clay to the laminated Stolleklint Clay at the onset of the CIE (Figure 6), this suggests a significantly higher accumulation rate during and after the CIE than before. The $\delta^{13}C$ curve observed during the body phase (-24.3 to 0 m) in the Stolleklint section is consistently stable at around -31‰ (Figure 5). The beginning of the recovery phase coincides with tephra layer #-33, one of the largest silicic layers within the whole sequence (Larsen et al., 2003), and is also marked by Hg/TOC anomalies (Figure 5). Sporadic tephras are observed into the Fur Formation, with the main phase of 140 ash layers (from #+1) beginning 27 m above tephra #-33 (Bøggild, 1918).

There are clear Hg/TOC anomalies that coincide with both the onset of the PETM and at the transition from the body to the recovery of the CIE. The measured Hg/TOC ratios and $\delta^{13}C$ values in the underlying Østerrende/Holmehus Clay are stable up until -24.64 m depth, where the first Hg/TOC anomaly occurs 0.07 m below the base of tephra layer 'SK1' (Figure 6). Hg/TOC ratios continue to fluctuate before and after the deposition of the two thick tephra layers 'SK1' and 'SK2' at the onset of the CIE. The overlying strata record extremely stable Hg/TOC ratios through the majority of the Stolleklint Clay, despite noticeable fluctuations in Hg and TOC concentrations (Figure 5). These stable Hg/TOC values correlate with an absence of visible tephra layers at this locality, and while the absence of tephra layers does not exclude volcanic activity at the time, it is noteworthy given the abundance of tephra layers above and below this interval at the onset and recovery of the CIE. Fluctuations away from this stable baseline only begin to occur above -5.6 m depth, just prior to the reemergence of preserved tephra layers in the form of ash #-39 from the numbered ash series (Bøggild, 1918).

### 4.4 Lomonosov Ridge, Arctic Ocean

The results from core 302-4A are shown in Figure 7. The discontinuous core recovery complicates the interpretation of Hg/TOC values, particularly as the onset of the CIE is missing and parts of the recovery period are absent due to insufficient remnant sample material. However, the shape of the preserved CIE, with a stable body followed by a recovery period (Sluijs et al., 2006), is consistent with continental shelf PETM sites. There is substantially less variation in Hg/TOC ratios through this core compared to the sites in Svalbard and Denmark, although this may result from insufficient data during key intervals. There are two anomalously high Hg/TOC values, one below the onset of the CIE and the other at the beginning of the recovery phase (Figure 7). Based on the data available, there do not appear to be significant Hg/TOC anomalies across the PETM.



### 4.5 Bass River, New Jersey

The results from the Bass River core are shown in Figure 8. This locality has low TOC concentrations (0.06-0.64 wt%) throughout the studied section, which leads to larger uncertainties in the Hg/TOC ratios. Nearly half of the samples have TOC concentrations <0.2 wt%, the threshold recommended by Grasby et al. (2016) for reporting Hg/TOC anomalies. These samples are omitted from the Hg/TOC graph in Figure 8. The variations in Hg/TOC values are considerably smaller than observed at the other sites, but due to the low TOC concentrations in the section this may not be reliable. Despite these limitations, the largest apparent Hg/TOC anomalies occur several meters below and above the CIE, although the unconformity at 347.05 m.b.s. means it is unclear how much later the clay succession was deposited after the PETM.

### 5 Discussion

The five sites investigated in this study display a wide variety of Hg/TOC ratios during the late Paleocene and early Eocene. The Grane core (North Sea) samples have Hg concentrations considerably greater than any previously published Hg anomalies in the rock record (Figure 3). The results from Svalbard, Fur, and Dababiya (Figures 4-6, 9) show pronounced Hg/TOC anomalies across the PETM that are comparable in magnitude to previously published records from other key intervals in Earth history (Font et al., 2016; Grasby et al., 2016; Sanei et al., 2012). Based on the data available, there is no evidence for clear Hg/TOC anomalies at the Lomonosov and Bass River localities (Figures 7-8). These results suggest that there is strong spatial variation in the recorded Hg/TOC anomalies and the strength of Hg/TOC anomalies is positively correlated with the locality's proximity to the NAIP.

The results from the study of the GSSP locality in Dababiya (Egypt) by Keller et al. (2018) are summarized in Figure 9. The results from the GSSP locality are hampered by several factors, including $\delta^{13}C_{carb}$ and $\delta^{13}C_{org}$ showing different shapes of the CIE (Khozyem et al., 2015), which makes it difficult to define various stages of the PETM. Moreover, the section is complicated by the limited number of biomarkers and organic microfossils (Speijer and Wagner, 2002) and the effects of *in situ* carbonate dissolution and weathering. This locality also suffers from the same problem as Bass River in that TOC concentrations are generally very low, with much of the Paleocene samples falling below the recommended threshold for reporting Hg/TOC values (Grasby et al., 2016). However, the section differs markedly from Bass River in that there are two distinct peaks in Hg concentrations occurring within the CIE (Figure 9). The first peak of 288 ppb occurs in a section that is heavily affected by carbonate dissolution, and is not matched by a concomitant increase in TOC values. The Hg/TOC anomalies appear extreme (up to 1090 ppb/wt%), although this value will have a considerable error due to the very low TOC concentrations. The second Hg peak occurs during a brief interval less affected by acidification, and once normalized to TOC there is no Hg/TOC anomaly (Figure 9).



There are a number of potential issues with the studied sections that need to be considered before any interpretation of Hg/TOC anomalies can be made.

## 5.1 Site-specific uncertainties

The extreme Hg concentrations measured throughout the Grane core indicate that something exceptional has occurred at this locality that may have overprinted the primary signal (Figure 3). An important consideration for North Sea localities is the post-depositional effects of oil and gas seepage as the Balder Formation is a known cap rock. Therefore, it is possible that the section has been affected by hydrocarbon migration. However, measurements of the oil sands at 1766.6 and 1783.7 m.b.s. had Hg concentrations of 1135 and 3295 ppb, respectively. Whilst Hg-rich, these values fall well short of the measured Hg concentrations before (36,750 ppb) and after (90,100 ppb) the PETM CIE (Figure 3). In the absence of a known process that would decouple Hg and C during hydrocarbon migration, we therefore interpret the most likely source of these extreme Hg concentrations as a proximal Hg source that was very localized in depositional extent. The most likely scenario for such a process is the dominant release of Hg to seawater, either through hydrothermal vent complexes and/or Surtseyan-type volcanic eruptions (Figure 2). The significant Hg/TOC anomalies throughout this section indicate that if the cause is syn-depositional, then significant Hg source(s) were active before, during, and after the PETM.

The Lomonosov section (Figure 7) is complicated by poor core recovery during the CIE onset and the unavailability of samples during the CIE recovery. Unfortunately, these missing intervals coincide with the largest Hg/TOC anomalies observed in Svalbard, Fur, and Dababiya. Therefore, the results from the Lomonosov core unlikely paint a fully representative picture. The Bass River and Dababiya localities (Figures 8-9) are affected by low TOC concentrations, which could cause large changes in Hg/TOC ratios that result from TOC variations within measurement uncertainties, hindering interpretation of any peaks (Grasby et al., 2016). As such, it is possible that small to medium Hg/TOC anomalies are lost to the background noise of the Bass River locality, and the uncertainty of the magnitude of the Hg/TOC anomaly at Dababiya is very high. It is conceivable, but unlikely, that the Hg/TOC anomaly at Dababiya could purely be a product of diagenetic and weathering processes, given the amount of dissolution and acidification observed at this site (Figure 9; Keller et al., 2018; Khozyem et al., 2015). However, the effects of such processes on Hg/TOC ratios are poorly understood.

## 5.2 Non-volcanic influences on Hg/TOC ratios

*Alternative Hg sources*

While volcanism is an established source of Hg to the environment, there are a number of other processes that could possibly lead to the rapid release of Hg, including those that have been suggested as possible causes for the PETM. The presence of impact ejecta at the start of the CIE has been used to suggest that a meteorite impact may have triggered the global warming (Schaller et al., 2016). Little is known about how a meteorite may affect the Hg cycle, but presumably the location of impact is key to any disturbance, as the affected sediments can have large variations in organic matter and therefore Hg concentrations.





Mercury concentrations in meteorites are believed to be low, based on studies of the Cretaceous–Paleogene boundary (Sial et al., 2013). Any possible Hg anomaly from volatilized impacted rocks would be manifest as a sharp, individual Hg/TOC peak in the rock record. There are several Hg/TOC peaks close to the onset of the PETM at several sites, but none that appear to correlate exactly at the CIE onset where the glass spherules have been observed (Schaller et al., 2016). Given that these records

have numerous distinct Hg/TOC peaks, and that chemical and isotopic records lack an extraterrestrial signal during this time (Schmitz et al., 2004), a meteorite impact can be discounted as a possible cause for the observed Hg/TOC anomalies.

Potential terrestrial Hg sources include permafrost thawing (DeConto et al., 2012; Schuster et al., 2018) or wild fires producing widespread fly-ash (Sanei et al., 2012). However, the late Paleocene was abnormally warm, with both poles ice-free and

temperatures much higher than present day at high latitudes (Bijl et al., 2009; Sluijs et al., 2006) due to the persistent polar amplification in these warm climates (Frieling et al., 2017), casting doubt as to whether significant permafrost reserves existed at this time. Wild fires can also be discounted as a cause of Hg/TOC anomalies, as increases in Hg loading from fly ash events appear to correlate with increased TOC deposition (Sanei et al., 2012). Moreover, organic matter during the CIE in Svalbard (Figure 4) is dominated by amorphous material and marine dinocysts (Harding et al., 2011), negating a fly-ash source of the

Hg/TOC anomalies. At present, there is no available data on marine methane hydrates and their potential role (if any) as a source of Hg to the environment.

*Other influences on Hg/TOC values*

The source of organic matter may also be an important factor for Hg/TOC values. Terrestrial and marine environments are

inherently different in terms of ecosystem dynamics and nutrient cycling, which may affect the degree of Hg enrichment in various organic carbon reservoirs on land and in the oceans. There is compelling evidence that the organic matter sources varied across the PETM for many localities. In Svalbard, palynological evidence indicates that there was a distinct transient shift towards marine-derived organic matter across the PETM (Harding et al., 2011). The organic matter before and after the CIE is dominated by terrestrially-derived phytoclasts of cuticle and wood, while the body of the PETM is largely comprised

of amorphous organic matter and marine dinocysts (Harding et al., 2011). The same trend is observed at the Lomonosov ridge, with abundant terrestrial organic matter observed before and after the CIE (Sluijs et al., 2006; Weller and Stein, 2008) and changes to salinity and runoff (Pagani et al., 2006; Sluijs et al., 2008; Waddell and Moore, 2008). Fluctuations in salinity and terrestrial runoff are also observed around the Bass River locality (Gibson et al., 2000; John et al., 2008; Zachos et al., 2006). It is therefore possible that a shift in the source of organic matter would lead to a change in the Hg/TOC baseline during the

PETM, such as that observed in Svalbard (Figure 3). However, a study of shelf sections during the emplacement of the Karoo–Ferrar LIP suggest that terrestrial organic matter generally has a higher Hg/TOC ratio than marine organic matter (Percival et al., 2015), implying that given the increased marine contribution to the TOC content, Hg/TOC values should decrease during the PETM, which is the opposite effect to that observed in the Svalbard core. The $\delta^{13}$C curve in Svalbard also has multiple transitional values between the onset and peak of the CIE, suggesting multiple admixed sources of organic matter (Harding et



al., 2011) that could mute a step change in Hg/TOC ratios. Indeed, the observed sea level rise across the PETM makes an increased marine contribution to TOC a likely factor at many shallow sites (e.g. Sluijs et al. 2008, Sluijs & Dickens 2012). While more work is needed to constrain Hg enrichments in various organic reservoirs (i.e. source specific Hg analyses) the source of organic matter does not appear to be a principal driver of Hg/TOC concentrations in these studied sections. Therefore,

if organic matter sourcing influenced Hg/TOC values, it is more likely to have muted Hg/TOC anomalies during the body of the CIE than to have caused them.

**5.3 NAIP magmatism as the cause of the Hg/TOC anomalies**

There are a number of lines of evidence that suggest that the emplacement of the NAIP is the prime candidate for the observed Hg/TOC anomalies. While there is considerable variation between localities, there are three discreet intervals at which Hg/TOC

anomalies most regularly occur: 1) just prior to the onset of the PETM, 2) during the recovery phase, and 3) later in the Eocene. These intervals also contain abundant tephra layers in the Grane, Fur, and, to a lesser extent, Svalbard localities (Figures 3-6), indicating that these Hg/TOC anomalies are likely to be volcanic/magmatic in origin. There are also elevated 'baseline' Hg/TOC values observed at Svalbard and Dababiya during the body of the CIE (Figures 4, 9).

*Pre-CIE onset*

The observed Hg/TOC anomalies prior to the onset of the CIE coincide with the first evidence of volcanism preserved after a significant hiatus in both Denmark (Heilmann-Clausen et al., 2014) and the North Sea (Haaland et al., 2000). At Fur, the stratigraphic level featuring the first rise of Hg/TOC also records numerous NAIP tephra beds (Figure 6), while at Grane Hg/TOC anomalies are present in the lowermost sample studied (Figure 3). In Svalbard, the numerous Hg/TOC anomalies

from around 5 m below the CIE onset are correlative with a shift to unradiogenic $^{187}Os/^{188}Os$ ratios (Wieczorek et al., 2013), which may indicate an enhanced flux of volcanically derived osmium to the ocean. This interval corresponds to around 18–23 kyr using the estimated CIE deposition rate (Charles et al., 2011). Such features are also observed from localities in the Arctic and Tethys Oceans prior to the onset of the CIE (Dickson et al., 2015; Schmitz et al., 2004). The temporal correlation of several volcanic proxies prior to the onset of the CIE suggests that the emplacement of the NAIP could be a primary factor driving

global warming prior to the PETM (Frieling et al., 2018; Sluijs et al., 2007b) and hence may function as a trigger for the CIE and extreme warming.

*CIE body*

The Hg/TOC results of the body of the CIE, defined as a period of stable negative $\delta^{13}C$ values, show significant variations

between the studied localities. In Svalbard, the consistently elevated Hg/TOC ratios during the body and into the recovery of the CIE suggest a prolonged source of Hg emissions that enhanced Hg deposition at this locality (Figure 4), possibly supported by the Dababiya section (Figure 9). For Grane, the analyzed core interval is swamped with extremely high Hg concentrations throughout (Figure 3). At Fur, there is no such enrichment in Hg, although this trend may be complicated by the sharp change



in lithology across the PETM interval (Figures 5-6). The Lomonosov and Bass River localities also show no evidence of increased Hg/TOC in the CIE body (Figures 7-8). The $^{187}Os/^{188}Os$ ratios during the CIE body in Svalbard lack a basaltic signature (Wieczorek et al., 2013), consistent with records from other ocean basins where the $^{187}Os/^{188}Os$ record is dominated by continental weathering (Dickson et al., 2015; Ravizza et al., 2001). Other volcanic proxies such as tephra layers are

uncommon at the Grane and Fur localities. Therefore, the sustained Hg/TOC increase observed in the Svalbard and Egypt sections appears regional in extent, and there is little other evidence for volcanism at that time.

*CIE Recovery*

There are numerous Hg/TOC anomalies preserved in the Fur and Svalbard sections close to the recovery of the PETM CIE.

This interval marks the beginning of hundreds of tephra layers deposited in Danish and North Sea sections during the early Eocene (Haaland et al., 2000; Larsen et al., 2003). The basaltic and silicic tephras in Denmark are >1200 km from the nearest known source volcano, indicative of the largest and most violent explosive eruptions known to occur (Mason et al., 2004). Individual explosive eruptions that deposited each Fur tephra layer are likely to have been 100's to 1000's km$^3$ in magma volume (Baines et al., 2008; Baines and Sparks, 2005), which is rare (~0.1 to 1 Myr recurrence time) for silicic volcanism and

unheard of for basaltic volcanism. These tephras are interpreted to have formed by shallow marine, phreatomagmatic eruptions that postdate both the opening of the northeast Atlantic Ocean (Planke et al., 2000) and the exposed east Greenland flood basalts (Storey et al., 2007b). It is therefore likely that the observed Hg/TOC anomalies in the early Eocene are caused by NAIP volcanism.

**5.4 NAIP magmatism as a cause for the PETM?**

The results presented here show significant Hg anomalies throughout the latest Paleocene and early Eocene, with the largest Hg/TOC anomalies occurring prior to the onset of the CIE and at the start of the recovery phase. The presence of abundant tephra layers at these intervals and the apparent variation in magnitude of Hg/TOC anomalies with proximity to the NAIP suggests that these anomalies are likely to represent volcanic and/or thermogenic eruptions releasing Hg to the environment. The temporal correlation with the onset of the CIE suggests that the emplacement of the NAIP could be a primary factor in

initiating the extreme global warming observed during the PETM. However, the relationship between the NAIP and global climate appears complex, as Hg/TOC anomalies and evidence for volcanism continue into the global cooling of the recovery phase and early Eocene. Moreover, the continuously elevated Hg/TOC concentrations observed during the CIE in Svalbard and Dababiya are less clearly volcanic in origin due to the absence of tephras and unradiogenic Os isotopes. Therefore, if the NAIP is responsible for initiating the PETM, then a combination of climate system feedbacks, varying magma production

rates, and/or the changes in the dominant method of magma emplacement are required to reconcile the available data.

The dominant emplacement style and magma production rates of the NAIP varied considerably during the Paleogene (Saunders, 2015; Storey et al., 2007b; Wilkinson et al., 2016), and potentially also across the PETM (Storey et al., 2007a).



One limitation is that volcanic proxies such as tephra deposition and Os isotopes may not reflect the true extent of magmatic activity, as they are dependent on the degree of explosive volcanism and basalt weathering, respectively. If the eruptions are dominantly effusive, or gas-dominated submarine eruptions from thermogenic sources, then tephra and Os isotope evidence may be more limited. Mercury anomalies may also be regionally limited by seawater interactions in the case of submarine

eruptions or hydrothermal venting (Figure 2), which may account for the extreme concentrations measured in the Grane core (Figure 3). Since volcanic eruptions and thermogenic gas release from contact metamorphism are inherently different processes, it is reasonable to assume that this is also reflected in mechanistic differences in C and Hg release (Figure 2). Therefore, the magnitude and spatial variations in Hg/TOC anomalies from these two sources may differ considerably, such that Hg/TOC anomalies do not necessarily always equate to the same C input. If the dominant mode of emplacement shifted

from volcanism to intrusions during the CIE, a mechanism proposed for other LIPs (Burgess et al., 2017), this could be consistent with more localized aqueous Hg dispersal and the release of $^{13}$C depleted carbon that could have maintained negative $\delta^{13}$C conditions (Frieling et al., 2016).

## 6 Conclusions

This study presents Hg/TOC anomalies across the PETM from five continental shelf sections. Extremely high Hg

concentrations are present in the Grane core in the North Sea and significant Hg/TOC anomalies are observed in Svalbard, Denmark, and Egypt (Keller et al., 2018). No such clear anomalies are observed in the Lomonosov Ridge or Bass River sections, although these localities are complicated by incomplete core recovery and low TOC concentrations, respectively. With exception of the Egypt section, the magnitude of Hg/TOC anomalies generally correlates with proximity to the NAIP, and the coincidence of other volcanic proxies indicates that the NAIP is the most likely source of the observed Hg/TOC

anomalies. The large variation in magnitudes of Hg/TOC anomalies between localities indicates that Hg deposition was regionally constrained. One possible explanation is that phreatomagmatic eruptions and submarine degassing from hydrothermal vent complexes meant greater transference of Hg to seawater, leading to localized deposition. Different organic matter sourcing may have muted Hg/TOC signals at some of these sites, but this effect is unlikely to cause positive Hg/TOC anomalies during the PETM.

There are two distinct disruptions to the Hg cycle, preserved in sections that are proximal to the NAIP. Tephra layers and Hg/TOC anomalies are most prevalent before the onset of the CIE and during the CIE recovery. In addition, the Svalbard section has Hg/TOC ratios that are consistently elevated during the CIE body. The most likely candidates for the observed Hg/TOC anomalies are the emplacement of an extensive flood basalt province along the margins of the nascent Northeast

Atlantic Ocean and the thermogenic gas release due to widespread sill intrusions. The anomalous Hg/TOC a few millennia prior to the CIE provides additional support that NAIP magmatism could have triggered a positive carbon cycle feedback, which released $^{13}$C-depleted carbon from a surface reservoir. Evidence for elevated magmatism at the end of the CIE, however,



highlights the relationship between postulated volcanic forcing and the climate is complex. Factors such as (sensitivity of) feedbacks in the climate system, a change in the emplacement style of the NAIP, and/or magma production rates may be key to both the onset and cessation of hyperthermal conditions during the PETM.

**Supplement link (will be included by Copernicus)**

Data Tables and Fur Island sampling description.

**Author contribution**

MTJ, LMEP, TAM, and HHS conceptualized and laid out the methodology of the project. MTJ, LMEP, EWS, JF, LR, BAS, BS, CT, SP, and HHS contributed to data collection and interpretations. Writing (original draft) was prepared by MTJ, LMEP, EWS, and JF. All authors contributed to the writing in the review and editing stage.

**Competing interests**

The authors declare that they have no conflict of interests.

**Acknowledgements**

Staff at Store Norske AS, Equinor ASA, Grace Shephard, Thea Heimdal, Mike Cassidy, Valentin Zuchuat, Olivia Jones, Claus Heilman-Clausen, Adriano Mazzini, Stephane Polteau, and Appy Sluijs are warmly thanked for their help and
assistance. This work supported by the Research Council of Norway through its Centers of Excellence funding scheme, project number 223272. MTJ and EWS are funded by the Research Council of Norway Yngeforsktalenter project 'Ashlantic', project number 263000.

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



**Figure 1:** A regional plate reconstruction at 56 Ma showing the sample sites used in this study. Each sample site is shown by a yellow star. Paleo-shorelines and marine seaways are from Golonka (2009). Present day coastlines are shown in black, while dark blue lines denote plate boundaries. Shelf areas are shown in light blue, while ocean basins in dark blue. The red areas show the known extent of subaerial and submarine extrusive volcanism of the North Atlantic Igneous Province (NAIP) at this time (Jones et al., 2016). The black areas show the known extent of NAIP sill intrusions in sedimentary basins (Planke et al., 2005; Rateau et al., 2013; Reynolds et al., 2017). Note that the identification of sills beneath extrusive volcanics is hampered by poor seismic retrievals. The figure was created with the help of Grace Shephard using the open source plate tectonics software GPlates (Boyden et al., 2011; Gurnis et al., 2012), based on modifications to the reconstruction model (Shephard et al., 2013) and plotted with Generic Mapping Tools (Wessel et al., 2013).





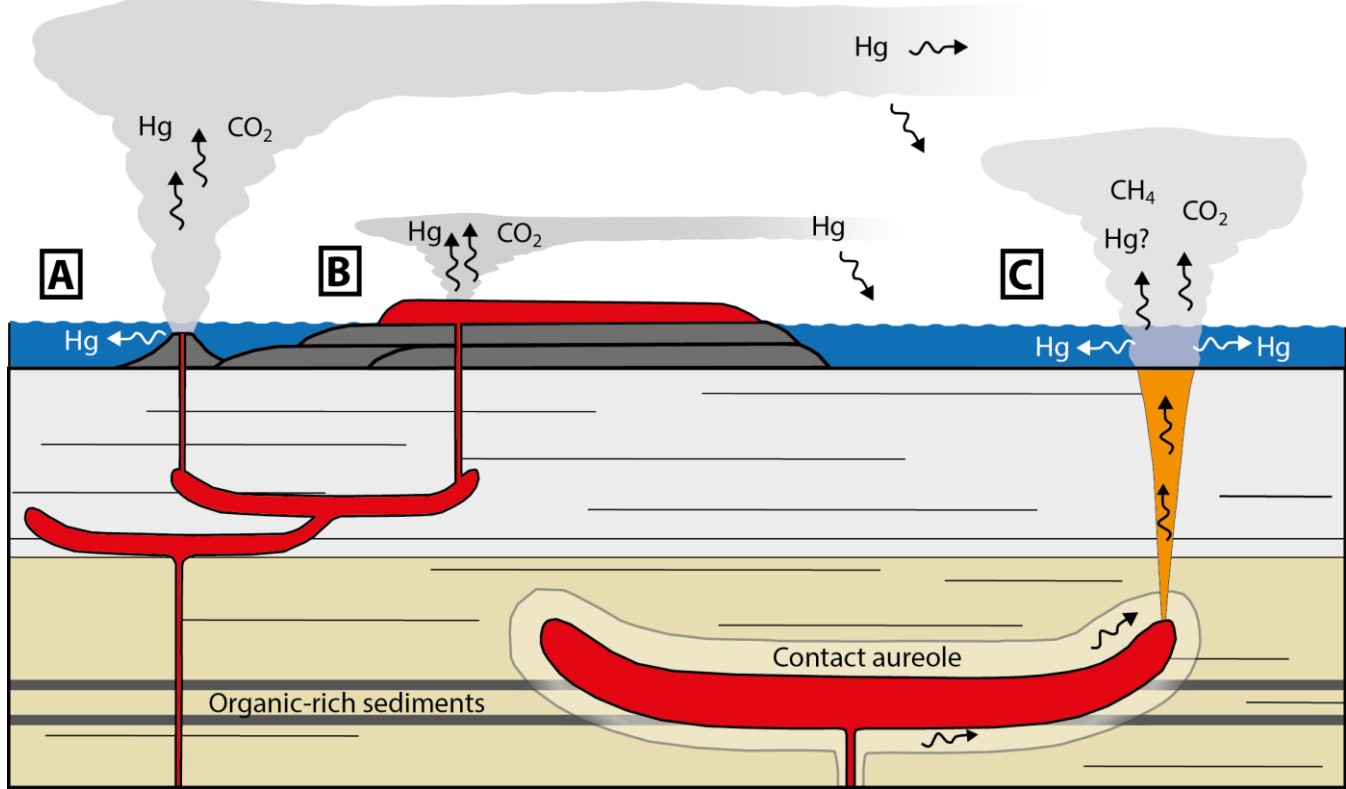

**Figure 2:** A schematic diagram (not to scale) showing the various emplacement methods of the NAIP and their control on the magnitude and distribution of Hg in the environment. A) Shallow marine eruptions lead to highly explosive Surtseyan-type volcanism, resulting in wide dispersal of tephra and Hg in the high atmosphere. Some Hg may be transferred to seawater. B) Effusive flood basalt volcanism releases Hg to the atmosphere, either the troposphere or stratosphere depending on the explosiveness of the eruption. C) Hydrothermal vent complex explosions from the edges of sill intrusions in organic-rich sedimentary basins. Mercury is likely to be added to the overlying water column, but some Hg may also reach the atmosphere.



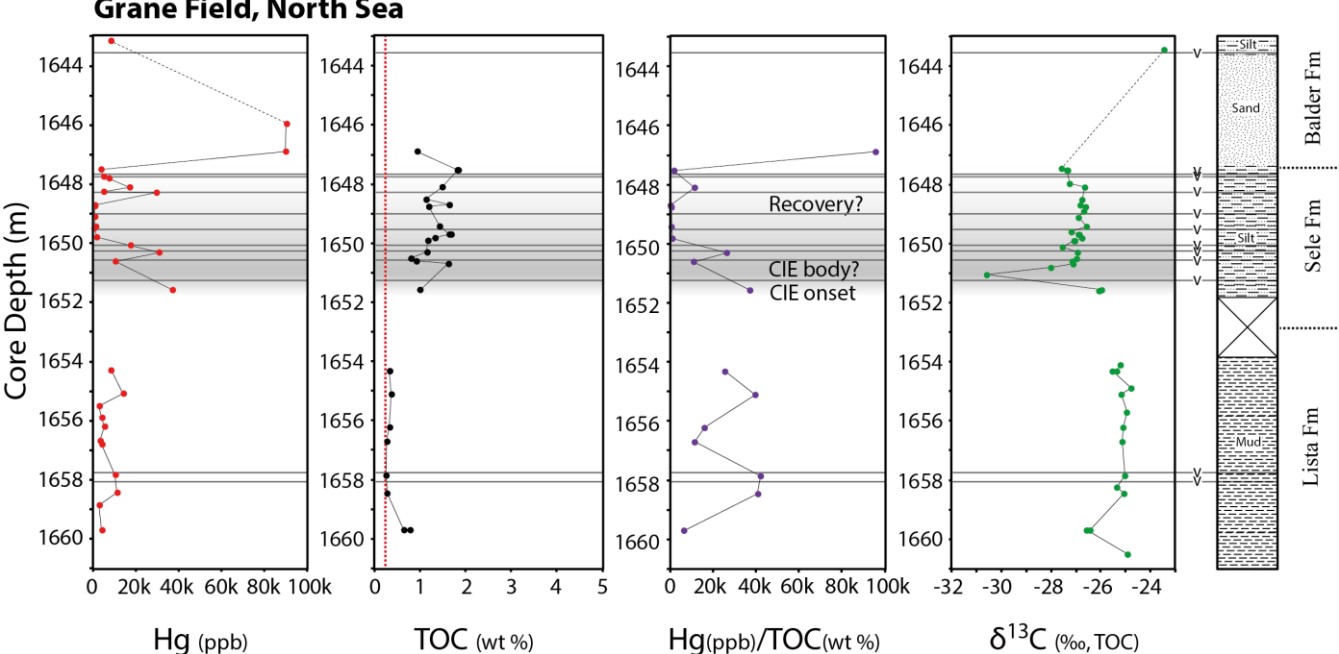

**Figure 3:** Mercury (Hg), total organic carbon (TOC), Hg/TOC, and $\delta^{13}C_{TOC}$ data from sections from core 25/11-17 from the Grane Field in the Norwegian North Sea. Mercury concentrations (ppb) are shown in red, total organic carbon (TOC) concentrations (wt %) in black, Hg/TOC ratios in purple, and $\delta^{13}C$ values in green. The stratigraphic log is adapted from Haaland et al. (2000). Incomplete core recovery is marked by crossed squares. The grey shaded area shows the approximate onset, body, and recovery phases of the carbon isotope excursion (CIE). Identified tephra layers are shown as dark grey bands across the figure. The red dashed line in the TOC graph denotes the recommended threshold for TOC concentrations to report Hg/TOC values (Grasby et al., 2016).



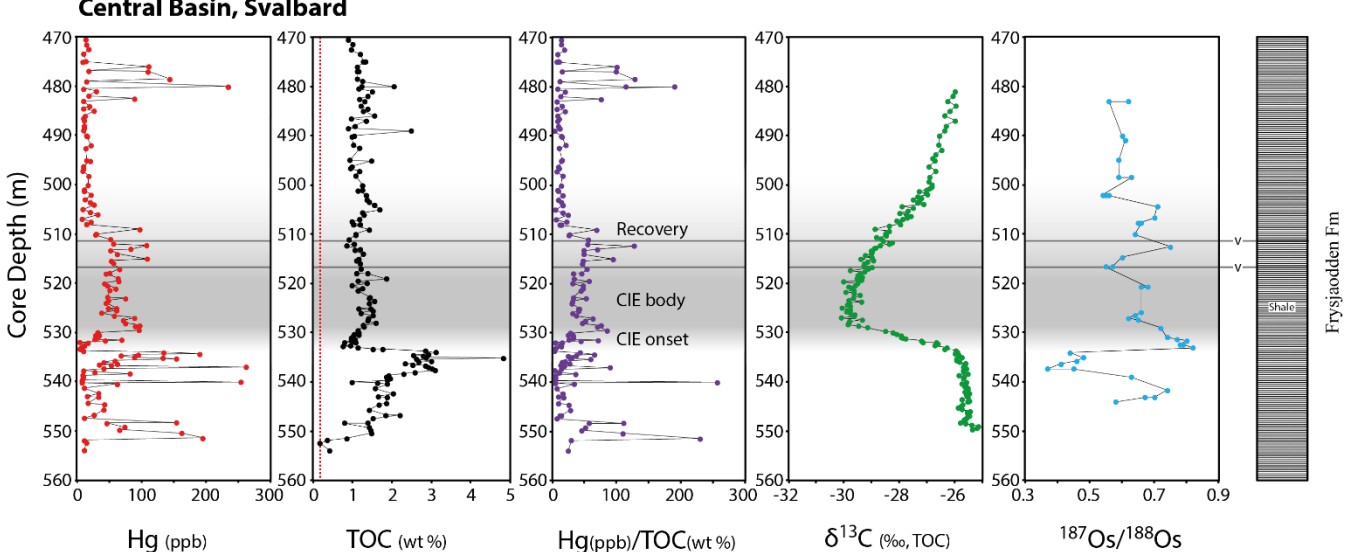

**Figure 4:** Mercury (Hg), TOC, Hg/TOC, $\delta^{13}C_{TOC}$, and $^{187}Os/^{188}Os$ data from sections from the BH09/2005 core from the Central Basin in Svalbard. The coloring system for Hg, TOC, Hg/TOC and $\delta^{13}C_{TOC}$ is the same as in Figure 3, while 187Os/188Os values are shown in light blue. The carbon isotope data and stratigraphic log are from Charles et al. (2011), while the Os isotope data is from Wieczorek et al. (2013). The grey shaded area shows the approximate onset, body, and recovery phases of the CIE. Identified tephra layers are shown as dark grey bands across the figure. The first of the two tephra layers has been radioisotopically dated to 55.785 ± 0.086 Ma (Charles et al., 2011). The red dashed line in the TOC graph denotes the recommended threshold for TOC concentrations to report Hg/TOC values (Grasby et al., 2016).




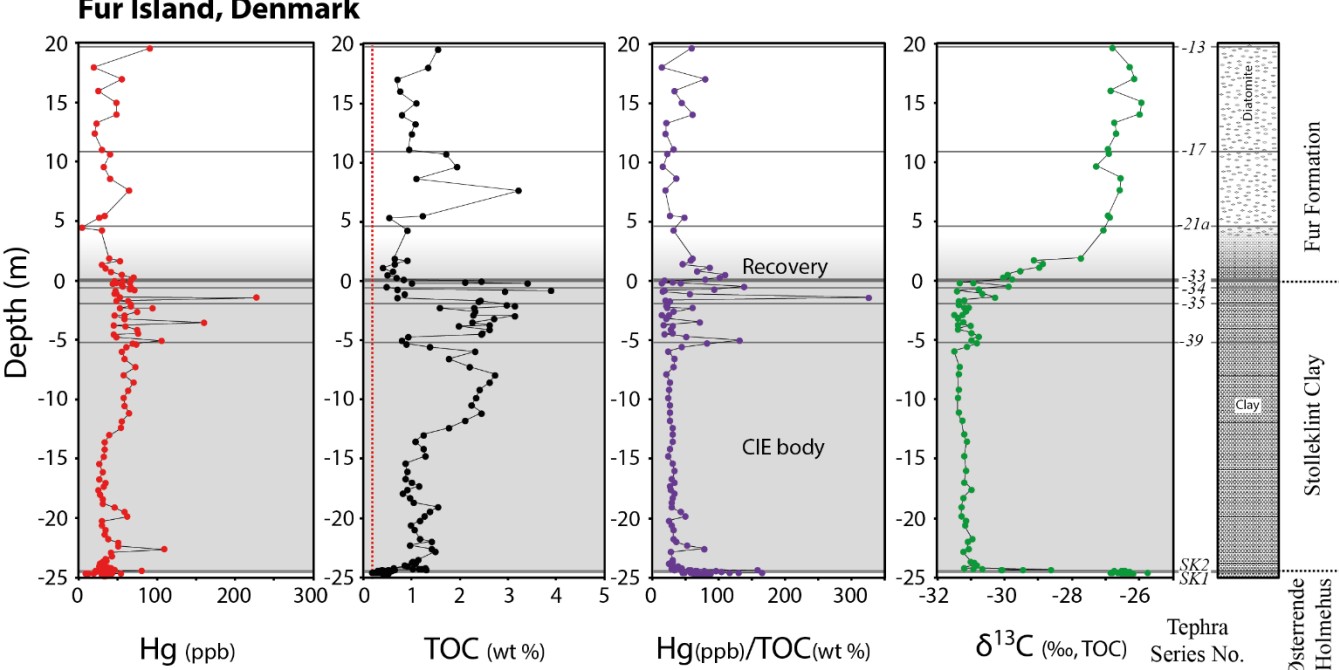

**Figure 5:** Mercury, TOC, Hg/TOC, and $\delta^{13}C_{TOC}$ data for the outcrops on Fur Island, Denmark. The coloring system is the same as in Figure 3. The depth estimates are calibrated using the tephra layers #-34 and #-33, with the top of tephra layer #-33 set to zero. See the Supplementary data for full details. Only the body and recovery phases of the CIE are marked due to the condensed nature of the onset phase (see Figure 6 for an expanded section). The stratigraphic log is based on the beach and cliff sections at Stolleklint. Boundaries between local formations are shown on the right (Heilmann-Clausen et al., 1985). Prominent tephra layers from the numbered ash series are marked with their corresponding number (Bøggild, 1918), along with the tephra layers 'SK1' and SK2' at the base of the Stolleklint Clay. The red dashed line in the TOC graph denotes the recommended threshold for TOC concentrations to report Hg/TOC values (Grasby et al., 2016).

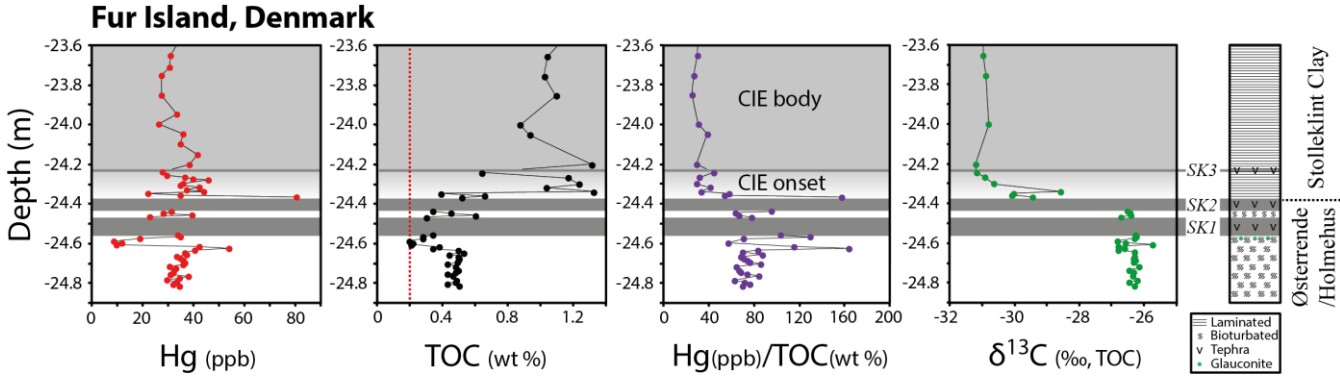

**Figure 6:** Mercury, TOC, Hg/TOC, and $\delta^{13}C_{TOC}$ data from an expanded section from the Stolleklint Beach section on Fur Island in Denmark, showing the onset of the CIE (see Figure 5). The coloring scheme is the same as in Figure 3. The grey shaded area shows the approximate onset and body phases of the CIE. Tephra layers 'SK1', SK2', and 'SK3' are marked at their respective depths, along with a stratigraphic log based on the beach section at Stolleklint. The y-axis scale is the calculated depth below the top of tephra layer #-33 (see supplementary data). The red dashed line in the TOC graph denotes the recommended threshold for TOC concentrations to report Hg/TOC values (Grasby et al., 2016). Samples with TOC values below this threshold were omitted from the Hg/TOC graph.



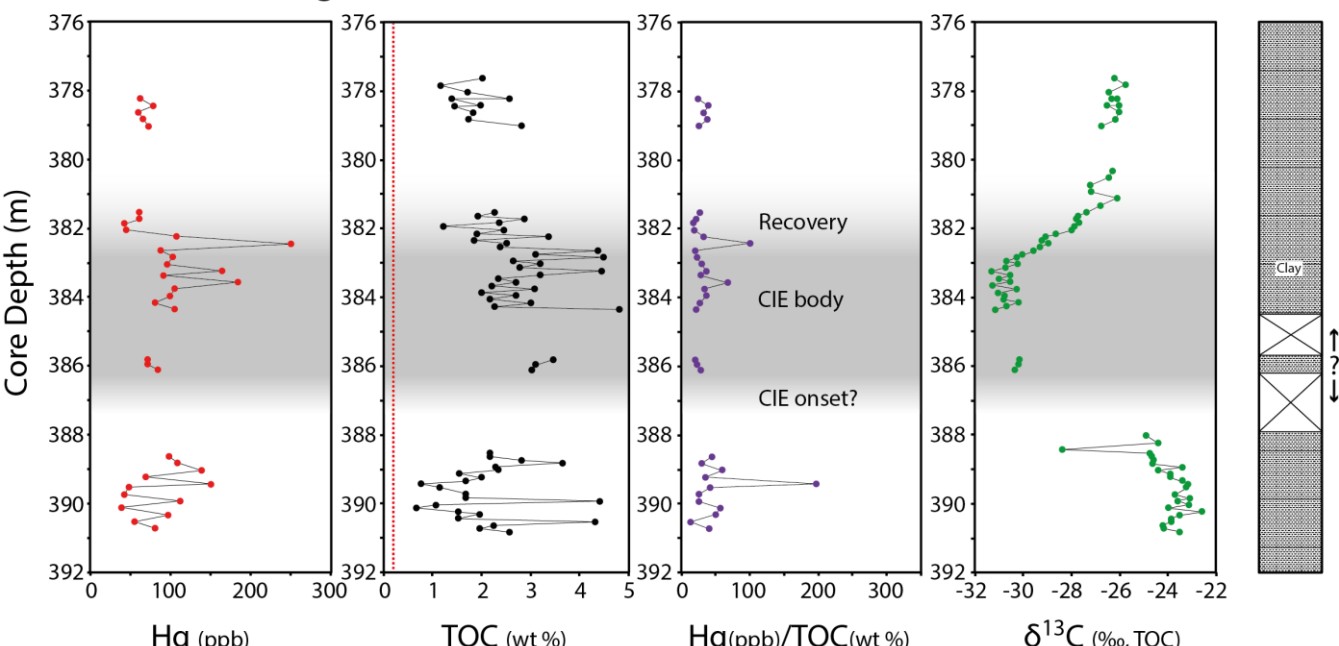

**Figure 7:** Mercury, TOC, Hg/TOC, and $\delta^{13}C_{TOC}$ data from the 302-4A core collected from the Lomonosov Ridge as part of the IODP Arctic Coring Expedition (ACEX). The coloring scheme is the same as in Figure 3. Incomplete core recovery is marked by crossed squares, with the middle fragment occurring at some unknown interval somewhere between the two sections of complete core recovery. The onset of the CIE is missing. The grey shaded area shows the approximate CIE body and recovery phases. Carbon isotope measurements are from Sluijs et al. (2006). The stratigraphic log is from Backman et al. (2004). The red dashed line in the TOC graph denotes the recommended threshold for TOC concentrations to report Hg/TOC values (Grasby et al., 2016).





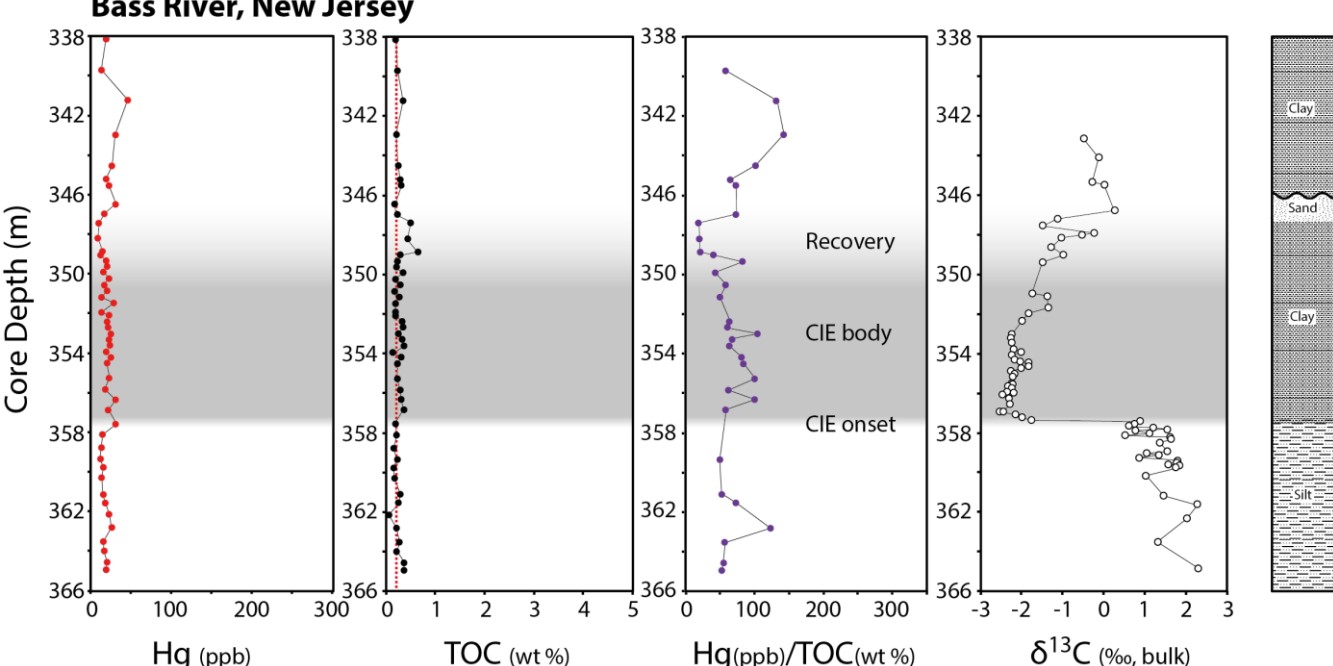

**Figure 8:** Mercury, TOC, Hg/TOC, and $\delta^{13}C_{bulk}$ data from the Ocean Drilling Project (ODP) leg 174 'Bass River' locality (Cramer et al., 1999). The coloring scheme is the same as in Figure 3, with the exception of $\delta^{13}C_{bulk}$, which is shown as white points. The $\delta^{13}C$ data and stratigraphic log are from John et al. (2008). An unconformity is found at 347.05 m.b.s. in the stratigraphy (Cramer et al., 1999). The grey shaded area shows the approximate onset, body, and recovery phases of the CIE. The red dashed line in the TOC graph denotes the recommended threshold for TOC concentrations to report Hg/TOC values (Grasby et al., 2016). Samples with TOC values below this threshold were omitted from the Hg/TOC graph.





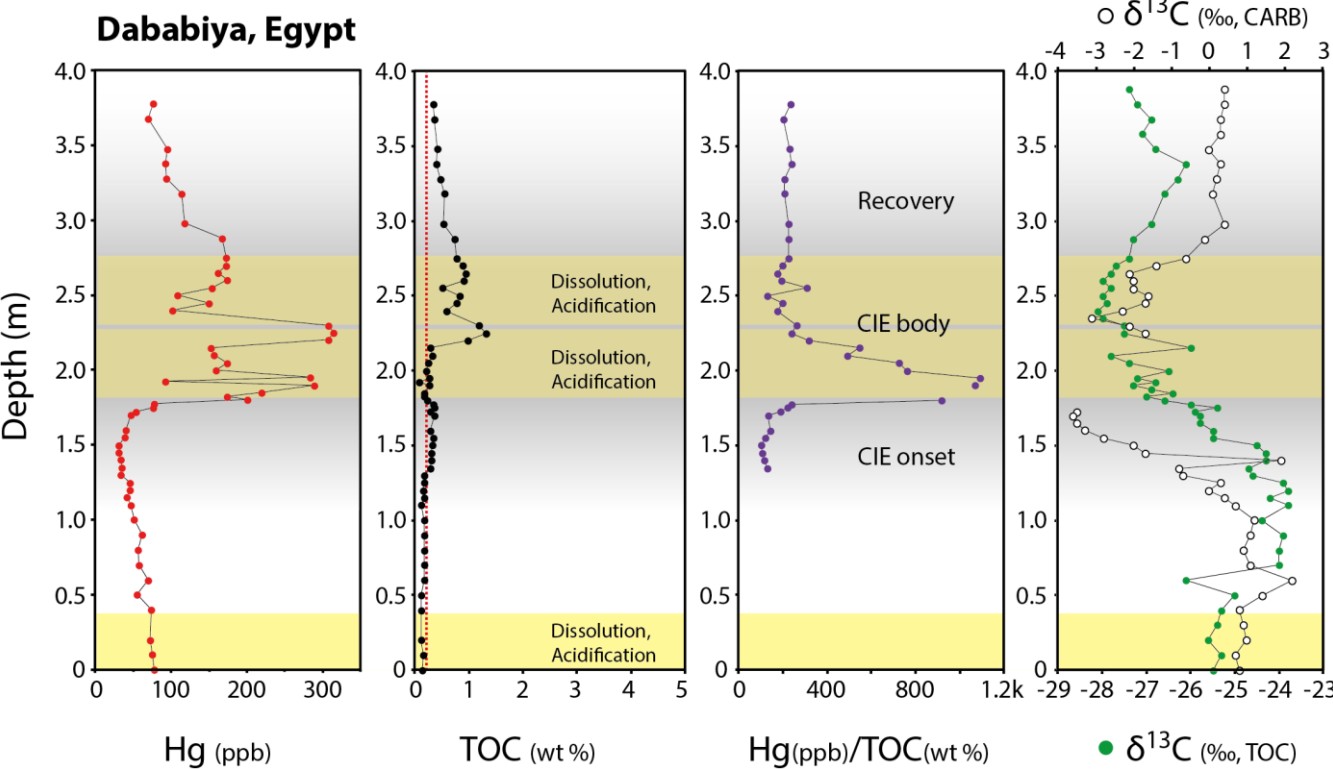

**Figure 9:** Mercury, TOC, Hg/TOC, $\delta^{13}C_{CARB}$ and $\delta^{13}C_{TOC}$ data from the Dababiya GSSP locality in Egypt (Keller et al., 2018). The coloring scheme is the same as in Figure 3, with the addition of $\delta^{13}C_{CARB}$ Data shown as white points. The red dashed line in the TOC graph denotes the recommended threshold for TOC concentrations to report Hg/TOC values (Grasby et al., 2016). Samples with TOC values below this threshold were omitted from the Hg/TOC graph. Yellow bands denote sections of stratigraphy heavily affected by dissolution and acidification (Keller et al., 2018).