# Peer review of "Mercury anomalies across the Paleocene-Eocene Thermal Maximum"

_Climate of the Past, 2018_

## Referee Comment (RC1) · Anonymous Referee #1 · 3 Dec 2018

Overall: This is a rich dataset aimed at investigating the link between NAIP volcanism (which ought to manifest as Hg/TOC anomalies in sediments) and the PETM. Purely based on the size of the dataset and the importance of the question being investigated, this is a worthwhile contribution. However, I do think that potential issues with the fidelity of individual records needs to be more fully acknowledged in the interpretation, which currently considers each Hg/TOC record to faithfully record volcanism. I find it entirely plausible that many of the anomalies at individual sites might be the result of secondary processes (changes in type of Corg, dissolution and diagenesis, even weathering of the outcrop sections) and not in fact related to volcanism at the time of deposition.

Abstract, Line 21: I assume the CIE you are referring to is the PETM, but that is not

clearly stated.

Page 3, Line 4-5: A "strong positive correlation" is a bit imprecise, I would rather state that a close temporal coincidence between several LIPS and mass extinctions has been noted.

Section 3.4: I suggest that the description of the PETM in the ACEX hole be ammended to reflect that the onset of the PETM is clearly missing, with the underlying 50cm (separated by a core gap) "disturbed by drilling and various proxies suggests that the sediment from this interval represents a mixture of uppermost Palaeocene and PETM material" (Sluijs et al, 2006).

Page 8, Line 8: "and represents the most distal locality to the volcanic activity of the NAIP studied here." Is it more distal than Dababiya?

Page 12 Line 232: "It is conceivable, but unlikely, that the Hg/TOC anomaly at Dababiya could purely be a product of diagenetic and weathering processes, given the amount of dissolution and acidification observed at this site (Figure 9; Keller et al., 2018; Khozyem et al., 2015). However, the effects of such processes on Hg/TOC ratios are poorly understood." Well, this begs the question then, could not the Hg/TOC anomalies (or lack thereof) at all of the sites be influenced by the effects of dissolution and diagenesis? All of the sites here have large changes in lithology across the PETM, which of course suggests variable susceptibility to diagenetic alteration, with Svalbard apparently the lone exception. Perhaps the authors should allow that the records at all other sites may be dominantly controlled by the changes in lithology, which preserve mercury and TOC to various degrees, while Svalbard (which actually has quite a convincing Hg anomaly coincident with the PETM), due to being fairly homogenous clay throughout, might be the most reliable record.

Page 13, Line 22: "In Svalbard, palynological evidence indicates that there was a distinct transient shift towards marine-derived organic matter across the PETM (Harding et al., 2011). The organic matter before and after the CIE is dominated by terrestriallyderived phytoclasts of cuticle and wood, while the body of the PETM is largely comprised of amorphous organic matter and marine dinocysts (Harding et al., 2011)." Well, that gives me second doubts about Svalbard being the most reliable record - if the nature of the organic matter (that hosts the Hg signal) is changing dramatically, then it's entirely possible that the trends preserved in the record are a result not of original Hg deposition, but how that Hg survives the ravages of time and diagenesis.

It seems to me that there's a potential problem here with the fidelity of the records. The effects of changes in the host organic material, subsequent dissolution, diagenesis and weathering on the preservation of Hg in sediments are all relatively unconstrained. It would be one thing if you had a set of sites of varying lithological changes, that all showed the same Hg/TOC trend. Then you could argue that the Hg/TOC trends are robust to diagenesis. But instead, I see a a number of sites with varying lithologic changes, that all show quite different trends in Hg/TOC. Now, I'm not suggesting that there's nothing to be learned, but perhaps the authors should stress that constraining the effects of all of those secondary processes is crucial to the interpretation of these Hg/TOC records. Before we have that information, it's difficult to make conclusions about NAIP volcanism and the PETM. These complications are discussed in sections 5.2, but not afterwards - in section 5.3 and in the conclusions the records are interpreted as is they are known to be robust. I find this to be an omission, and suggest adding statements to the effect of: Hg/TOC anomalies at individual sites (or the lack thereof) may reflect changes in the way Hg is preserved in sediments, not related to NAIP volcanism.

---

## Referee Comment (RC2) · Anonymous Referee #2 · 21 Dec 2018

**GENERAL COMMENTS**

This paper investigates Hg concentrations and Hg/TOC ratios in 6 continental shelf sections that span the PETM. The presence of Hg anomalies and the architecture of the Hg and Hg/TOC curves varies substantially among sites. The authors attribute these variations primarily to site location: sites closer the NAIP contain Hg anomalies while sites that are more distant do not. Where anomalies are present, they appear prior to the onset of the CIE and/or during the recovery phase. An exception is Svalbard, where there is also a sustained anomaly during the body of the CIE. The authors interpret the presence of Hg anomalies prior to the onset of the CIE as evidence that the NAIP triggered the initial warming during the PETM. They also suggest that more localized deposition of Hg is consistent with substantial Hg release by submarine hydrothermal

vents.

Overall, this is a well-written and thoughtful paper that presents a large data set relevant to unravelling the relationship between the NAIP and the PETM. One particular strength of this paper is its relatively detailed explanations of some of the uncertainties regarding the use of Hg/TOC curves as proxies for volcanism (in general) and site-specific uncertainties regarding the interpretation of the Hg/TOC records presented in this study. In my view, there has not been enough critical evaluation of the potential problems with applying Hg or Hg/TOC as a proxy for LIP volcanism in much of the recent literature.

That said, despite the detailed articulation of such uncertainties, the authors largely discount the influence of potentially complicating factors (e.g. influence of oil/gas migration, dissolution, weathering, diagenesis) in their interpretation of the PETM Hg and Hg/TOC records. I think the paper would be strengthened by giving greater consideration to the possibility that anomalies in some of these sections may not directly record volcanism. Thus, the authors could use this data set not only to explore the role of the NAIP in triggering the PETM, but also to emphasize the complexities associated with using Hg/TOC as a proxy for volcanism. To be clear, I am not suggesting fundamentally changing the conclusions of the paper, but instead giving more weight to alternative explanations for some of the Hg anomalies (particularly in the Grane Field and at the Dababiya locality).

SPECIFIC COMMENTS

-The extraordinarily high Hg concentrations in the Grane Field section are certainly worthy of attention, but difficult to interpret due to the potential effects of oil and gas seepage. Despite the possible effects of hydrocarbon migration, the authors conclude that high Hg concentrations before/after the CIE are likely due to Hg release by hydrothermal vent complexes associated with the NAIP. They base this conclusion on two lines of reasoning: 1) sediments before/after the CIE have higher Hg concentrations (by orders of magnitude) than the oil sands lower in the section, and 2) there are no known processes that can decouple Hg and organic carbon in hydrocarbon systems. However, as far as I'm aware, there is very little research about Hg reservoirs in hydrocarbon systems or Hg loss/gain during hydrocarbon maturation/migration. Thus, it seems premature to discount the role of hydrocarbon systems in generating these unusual values.

-The evidence for dissolution/weathering during the body of the CIE at the Dababiya, Egypt locale makes interpretation of the Hg record difficult. It is not clear on what basis the authors suggest that it is "unlikely" that the Hg/TOC anomaly at Dababiya is purely a product of diagenetic and weathering processes. Since "the effects of such processes on Hg/TOC ratios are poorly understood," wouldn't it be prudent to reserve judgement?

ADDITIONAL CITATIONS

The authors may want to incorporate findings from a recent paper that investigated the Hg isotopic composition of PETM and Eocene sediments from Lomonosov Ridge in the Arctic (Gleason et al., 2017, Sources and cycling to mercury in the paleo Arctic Ocean from Hg stable isotope variations in Eocene and Quaternary sediments: Geochimica and Cosmochimica Acta 197:245-262.). In short, this paper found that sediment from the PETM had a Hg isotopic composition consistent with that of Holocene sediments from the Arctic Ocean. This supports the conclusion that there was no large perturbation to the Hg cycle at this locality during the PETM.

LINE COMMENTS

Page 3, line 4: consider "temporal association" instead of "strong positive correlation"

page 3, lines 13-14: Although there is a need for "a well-tested and uniquely volcanic tracer in sedimentary rocks" – Hg anomalies are not unique tracers of volcanism. Hg anomalies could theoretically be generated through many different processes that release Hg – wildfires, permafrost thawing, meteorite impacts, etc. This sentence seems to foreshadow the Hg/TOC ratios as a uniquely volcanic tracer – which gives the wrong impression.

Page 3, line 19: Normalizing to TOC accounts for changes in Hg due to changes in the drawdown of organic carbon; it doesn't necessarily account for changes in sediment accumulation rate.

Page 3, lines 20-21: - "proposed' instead of "reported"

Page 3, lines 23-24: Rather than say "therefore, this method is an important proxy," consider, "therefore, we use this method as proxy" for volcanism. Given the potential for mercury anomalies to reflect processes/sources other than volcanism (as acknowledged in the paper and detailed above), I would be more careful with language here.

---

## Editor Comment (EC1) · Reyes (Editor) · 28 Dec 2018

Dear Dr. Jones and co-authors,

The discussion period for your manuscript is now over, and two reviewers have posted comments. Both reviewers appreciated the depth of the dataset you compiled, and agreed that the underlying research question is valuable. There was also consensus that your manuscript would be strengthened by a more critical assessment of the individual records in your compilation, in terms of potential non-volcanic factors that complicate interpretation of the individual Hg and Hg/TOC records.

Please respond to the reviewer comments in the online Discussion forum. If you make any changes to your manuscript, please clearly indicate the nature of these changes

in your response. Once I have reviewed your response, I fully anticipate inviting you to submit a revised manuscript for a final decision.

Sincerely, Alberto Reyes

---

## Author Response (AR1)

**Response to Reviewers:**

Reviewer comments in plain text, replies in italics

**5 Reviewer 1**

Overall: This is a rich dataset aimed at investigating the link between NAIP volcanism (which ought to manifest as Hg/TOC anomalies in sediments) and the PETM. Purely based on the size of the dataset and the importance of the question being investigated, this is a worthwhile contribution. However, I do think that potential issues with the fidelity of individual records needs to be more fully acknowledged in the interpretation, which currently considers each Hg/TOC record to faithfully record

10 volcanism. I find it entirely plausible that many of the anomalies at individual sites might be the result of secondary processes (changes in type of Corg, dissolution and diagenesis, even weathering of the outcrop sections) and not in fact related to volcanism at the time of deposition.

We thank the reviewer for this thorough and constructive review. We acknowledge that the manuscript can be improved in terms of clarity, particularly regarding the uncertainties introduced by post-depositional processes. To that end, we have made

15 edits to the discussion and conclusion, further clarifying current limitations of the dataset and identify important unknowns to move the field of Hg as a volcanic proxy forward.

Abstract, Line 21: I assume the CIE you are referring to is the PETM, but that is not clearly stated. *We have added 'PETM' to make that clear in the text*

20

25

Page 3, Line 4-5: A "strong positive correlation" is a bit imprecise, I would rather state that a close temporal coincidence between several LIPS and mass extinctions has been noted.

We have adapted the text based on the reviewer's recommendation. The sentence now reads: "There is a close temporal coincidence between the emplacement of LIPs and both rapid climate change events and mass extinctions in Earth history (Courtillot and Renne, 2003; Wignall, 2001), suggesting a possible causal connection."

Section 3.4: I suggest that the description of the PETM in the ACEX hole be amended to reflect that the onset of the PETM is clearly missing, with the underlying 50cm (separated by a core gap) "disturbed by drilling and various proxies suggests that the sediment from this interval represents a mixture of uppermost Palaeocene and PETM material" (Sluijs et al, 2006).

30 We have amended the text accordingly. The sentences now read: "Core recovery of PETM strata was poor, with the onset of the PETM completely missing. Only 55 cm of disturbed core (302-31X) was recovered from anywhere between 388 meters composite depth (m.c.d., the top of core 302-32X) and 384.54 m.c.d. (the bottom of core 302-30X). The core material was disturbed during drilling and this interval represents a mixture of uppermost Paleocene and PETM material (Sluijs et al., 2006)."

Page 8, Line 8: "and represents the most distal locality to the volcanic activity of the NAIP studied here." Is it more distal than Dababiya?

Dababiya is more distal than the Bass River locality, but the Dababiya is not one of our sections. We have amended the text accordingly to draw the attention of the reader to the fact. The text now reads: "The site represents the most distal locality to the undernie activity of the NAIB studied here, although the requirements and Dababiya activity of the 2018) is more

40 the volcanic activity of the NAIP studied here, although the previously studied Dababiya section (Keller et al., 2018) is more distal."

Page 12 Line 23: "It is conceivable, but unlikely, that the Hg/TOC anomaly at Dababiya could purely be a product of diagenetic and weathering processes, given the amount of dissolution and acidification observed at this site (Figure 9; Keller et al., 2018;

45 Khozyem et al., 2015). However, the effects of such processes on Hg/TOC ratios are poorly understood." Well, this begs the question then, could not the Hg/TOC anomalies (or lack thereof) at all of the sites be influenced by the effects of dissolution and diagenesis?

The potential role of weathering is significantly different between the Dababiya section and the sites studied here. Four of the five localities are cored sections of marine sediments, which means it is very unlikely there has been any subaerial weathering. The Fur Island section is a rapidly grading cliff face, retreating at a rate of approximately 0.5-1 m per year. Soil formation on

50 The Fur Island section is a rapidly eroding cliff face, retreating at a rate of approximately 0.5-1 m per year. Soil formation on

35

top of the section is also limited and we argue the chances of significant surface weathering are very low. Although the degree of thermal maturation changes from immature (Fur, Bass River, ACEX) to somewhat mature (Svalbard, Grane) for all our localities, organic matter is well preserved. In contrast, the Dababiya section and other sites in Egypt are characterised by very degraded organic matter, signalling alteration of the OM through weathering and/or excessive heating (e.g. Speijer &

5 Wagner, 2002). We further clarified this important difference in our revised text, adding a paragraph at the beginning of the section 5.2 – "Other influences on Hg/TOC values."

All of the sites here have large changes in lithology across the PETM, which of course suggests variable susceptibility to diagenetic alteration, with Svalbard apparently the lone exception. Perhaps the authors should allow that the records at all other

10 sites may be dominantly controlled by the changes in lithology, which preserve mercury and TOC to various degrees, while Svalbard (which actually has quite a convincing Hg anomaly coincident with the PETM), due to being fairly homogenous clay throughout, might be the most reliable record.

In terms of diagenesis, we cannot completely exclude the possibility of these processes affecting these sections, with possible knock-on effects to the resulting Hg and Hg/TOC signals. The theory the reviewer proposes seems entirely plausible for some of the signals we observe (such as the broad Hg/TOC increase during the PETM CIE at Svalbard), However, given recent evidence (Them et al., 2019), we would in fact expect an anomaly in the other direction. We also emphasize it is difficult to reconcile variable diagenetic effects and the sharp Hg/TOC anomalies within homogenous lithological layers, such as we observe at Fur and Svalbard. We have added a paragraph to the discussion (section 5.2 – "Other influences on Hg/TOC values") to discuss this further, and included the recent work by Them et al (2019), which considers these problems in more

20 detail.

Page 13, Line 22: "In Svalbard, palynological evidence indicates that there was a distinct transient shift towards marine-derived organic matter across the PETM (Harding et al., 2011). The organic matter before and after the CIE is dominated by terrestrially-derived phytoclasts of cuticle and wood, while the body of the PETM is largely comprised of amorphous organic

- 25 matter and marine dinocysts (Harding et al., 2011)." Well, that gives me second doubts about Svalbard being the most reliable record - if the nature of the organic matter (that hosts the Hg signal) is changing dramatically, then it's entirely possible that the trends preserved in the record are a result not of original Hg deposition, but how that Hg survives the ravages of time and diagenesis. It seems to me that there's a potential problem here with the fidelity of the records. The effects of changes in the host organic material, subsequent dissolution, diagenesis and weathering on the preservation of Hg in sediments are all
- 30 relatively unconstrained. It would be one thing if you had a set of sites of varying lithological changes, that all showed the same Hg/TOC trend. Then you could argue that the Hg/TOC trends are robust to diagenesis. But instead, I see a number of sites with varying lithologic changes, that all show quite different trends in Hg/TOC. Now, I'm not suggesting that there's nothing to be learned, but perhaps the authors should stress that constraining the effects of all of those secondary processes is crucial to the interpretation of these Hg/TOC records. Before we have that information, it's difficult to make conclusions about
- 35 NAIP volcanism and the PETM. These complications are discussed in sections 5.2, but not afterwards in section 5.3 and in the conclusions the records are interpreted as is they are known to be robust. I find this to be an omission, and suggest adding statements to the effect of: Hg/TOC anomalies at individual sites (or the lack thereof) may reflect changes in the way Hg is preserved in sediments, not related to NAIP volcanism.

As mentioned above, we have added to the discussion (5.2) that adds more clarity to the relative importance of various 40 unknowns in the Hg cycle, and what this means in terms of limitations to the conclusions that can be drawn from this study. We have also added sentences to the Abstract, section 5.4, and the Conclusion to address these uncertainties.

**Reviewer 2**

**GENERAL COMMENTS**

- 45 This paper investigates Hg concentrations and Hg/TOC ratios in 6 continental shelf sections that span the PETM. The presence of Hg anomalies and the architecture of the Hg and Hg/TOC curves varies substantially among sites. The authors attribute these variations primarily to site location: sites closer the NAIP contain Hg anomalies while sites that are more distant do not. Where anomalies are present, they appear prior to the onset of the CIE and/or during the recovery phase. An exception is Svalbard, where there is also a sustained anomaly during the body of the CIE. The authors interpret the presence of Hg
- 50 anomalies prior to the onset of the CIE as evidence that the NAIP triggered the initial warming during the PETM. They also

suggest that more localized deposition of Hg is consistent with substantial Hg release by submarine hydrothermal vents. Overall, this is a well-written and thoughtful paper that presents a large data set relevant to unravelling the relationship between the NAIP and the PETM. One particular strength of this paper is its relatively detailed explanations of some of the uncertainties regarding the use of Hg/TOC curves as proxies for volcanism (in general) and site specific uncertainties regarding the intermetation of the Ug/TOC magnetic grade grade the study. In my view, there has not have been explored in this study of the study of the set of t

5 interpretation of the Hg/TOC records presented in this study. In my view, there has not been enough critical evaluation of the potential problems with applying Hg or Hg/TOC as a proxy for LIP volcanism in much of the recent literature.

That said, despite the detailed articulation of such uncertainties, the authors largely discount the influence of potentially complicating factors (e.g. influence of oil/gas migration, dissolution, weathering, diagenesis) in their interpretation of the

10 PETM Hg and Hg/TOC records. I think the paper would be strengthened by giving greater consideration to the possibility that anomalies in some of these sections may not directly record volcanism. Thus, the authors could use this data set not only to explore the role of the NAIP in triggering the PETM, but also to emphasize the complexities associated with using Hg/TOC as a proxy for volcanism. To be clear, I am not suggesting fundamentally changing the conclusions of the paper, but instead giving more weight to alternative explanations for some of the Hg anomalies (particularly in the Grane Field and at the

**15 Dababiya locality).**

We also thank the second reviewer for their detailed comments on the manuscript. As with the previous reviewer, we recognise that certain parts of the manuscript can be improved in terms of clarity and recognition of uncertainties when attributing Hg anomalies to volcanic activity. We have addressed all of the specific comments below, and made small edits throughout the manuscript to improve the critical evaluation of this dataset.

**20**

**SPECIFIC COMMENTS**

The extraordinarily high Hg concentrations in the Grane Field section are certainly worthy of attention, but difficult to interpret due to the potential effects of oil and gas seepage. Despite the possible effects of hydrocarbon migration, the authors conclude that high Hg concentrations before/after the CIE are likely due to Hg release by hydrothermal vent complexes associated with

- 25 the NAIP. They base this conclusion on two lines of reasoning: 1) sediments before/after the CIE have higher Hg concentrations (by orders of magnitude) than the oil sands lower in the section, and 2) there are no known processes that can decouple Hg and organic carbon in hydrocarbon systems. However, as far as I'm aware, there is very little research about Hg reservoirs in hydrocarbon systems or Hg loss/gain during hydrocarbon maturation/migration. Thus, it seems premature to discount the role of hydrocarbon systems in generating these unusual values.
- 30 We recognise that it is too early to be able to conclusively tie the Grane field Hg anomalies to volcanic activity due to the uncertainties associated with hydrocarbon migration, although given that decoupling of Hg and C usually occurs during volatilization, it is difficult to reconcile the lower Hg concentrations in the oil sands with the strata above. We have edited the text to draw the attention of the reader to these uncertainties and are more cautious in our interpretations. The sentences in section 5.1 now read: "We cannot discount the possibility that Hg and C could become decoupled during hydrocarbon
- 35 migration, although how this might occur is not known as decoupling usually takes place during volatilization. It is our view that the most plausible hypothesis for these extreme Hg concentrations is a proximal Hg source that was very localized in depositional extent." We have also made small edits throughout the manuscript to address the reviewer's concerns.

The evidence for dissolution/weathering during the body of the CIE at the Dababiya, Egypt locale makes interpretation of the

40 Hg record difficult. It is not clear on what basis the authors suggest that it is "unlikely" that the Hg/TOC anomaly at Dababiya is purely a product of diagenetic and weathering processes. Since "the effects of such processes on Hg/TOC ratios are poorly understood," wouldn't it be prudent to reserve judgement?

We agree, and have altered the text accordingly. The text at the end of section 5.1 now reads: "It is conceivable that the Hg/TOC anomalies at Dababiya could purely be a product of diagenetic and weathering processes, given

45 the amount of dissolution and acidification observed at this site (Figure 9; Keller et al., 2018; Khozyem et al., 2015). However, the effects of such processes on Hg/TOC ratios are poorly understood."

**ADDITIONAL CITATIONS**

The authors may want to incorporate findings from a recent paper that investigated the Hg isotopic composition of PETM and 50 Eocene sediments from Lomonosov Ridge in the Arctic (Gleason et al., 2017, Sources and cycling to mercury in the paleo Arctic Ocean from Hg stable isotope variations in Eocene and Quaternary sediments: Geochimica and Cosmochimica Acta 197:245-262.). In short, this paper found that sediment from the PETM had a Hg isotopic composition consistent with that of Holocene sediments from the Arctic Ocean. This supports the conclusion that there was no large perturbation to the Hg cycle at this locality during the PETM.

5 We thank the reviewer for bringing this paper to our attention. We have added a sentence to the first paragraph of the discussion, which reads: "A recent study on PETM sediments from the Lomonosov Ridge found that Hg isotopes across this interval were comparable to that of Holocene sediments (Gleason et al., 2017), supporting the conclusion that there was no large perturbation to the Hg cycle at this locality during the PETM in the available strata."

**10 LINE COMMENTS**

Page 3, line 4: consider "temporal association" instead of "strong positive correlation" As with the suggestion by reviewer 1, we have adapted the text based on the reviewer's recommendation. The sentence now reads: "There is a close temporal coincidence between the emplacement of LIPs and both rapid climate change events and mass extinctions in Earth history (Courtillot and Renne, 2003; Wignall, 2001), suggesting a possible causal connection."

15

25

35

page 3, lines 13-14: Although there is a need for "a well-tested and uniquely volcanic tracer in sedimentary rocks" – Hg anomalies are not unique tracers of volcanism. Hg anomalies could theoretically be generated through many different processes that release Hg – wildfires, permafrost thawing, meteorite impacts, etc. This sentence seems to foreshadow the Hg/TOC ratios as a uniquely volcanic tracer – which gives the wrong impression.

20 To omit any foreshadowing, we have removed part of the text suggesting that this would be a uniquely volcanic tracer. The sentence now reads: "Therefore, a well-tested volcanic tracer in sedimentary rocks would be a powerful proxy for understanding the temporal relationship between large-scale volcanism and rapid climate change events.

Page 3, line 19: Normalizing to TOC accounts for changes in Hg due to changes in the drawdown of organic carbon; it doesn't necessarily account for changes in sediment accumulation rate.

We have change the text accordingly. The sentence now reads: "Normalizing Hg to Hg/TOC helps to correct for variations in the drawdown of organic carbon, as organic compounds are generally the primary phases to complex Hg (Sanei et al., 2012)."

Page 3, lines 20-21: - "proposed' instead of "reported"

30 We have change the text accordingly. The sentence now reads: "Such anomalies have been proposed as a tracer for volcanism for several major environmental perturbations and/or mass extinctions in the geological record."

Page 3, lines 23-24: Rather than say "therefore, this method is an important proxy," consider, "therefore, we use this method as proxy" for volcanism. Given the potential for mercury anomalies to reflect processes/sources other than volcanism (as acknowledged in the paper and detailed above), I would be more careful with language here.

We have change the text accordingly. The sentence now reads: "Therefore, we use this method as a proxy to assess the relative importance of volcanism as a causal mechanism for the PETM, given the wealth of literature on the PETM and the availability of numerous sedimentary sections worldwide."

**Mercury anomalies across the Palaeocene-Eocene Thermal Maximum**

Morgan T. Jones1, Lawrence M.E. Percival2†, Ella W. Stokke1, Joost Frieling3, Tamsin A. Mather2, Lars Riber4, Brian A. Schubert5, Bo Schultz6, Christian Tegner7, Sverre Planke1,8, Henrik H. Svensen1

[revised manuscript text omitted]